# OxyS small RNA induces cell cycle arrest to allow DNA damage repair

Shir Barshishat[1,†], Maya Elgrably-Weiss[1,†], Jonathan Edelstein[1,†], Jens Georg[2,†], Sutharsan Govindarajan[1], Meytal Haviv[1], Patrick R Wright[3], Wolfgang R Hess[2,*] & Shoshy Altuvia[1,**]

## Abstract

**To maintain genome integrity, organisms employ DNA damage response, the underlying principles of which are conserved from bacteria to humans. The bacterial small RNA OxyS of *Escherichia coli* is induced upon oxidative stress and has been implicated in protecting cells from DNA damage; however, the mechanism by which OxyS confers genome stability remained unknown. Here, we revealed an OxyS-induced molecular checkpoint relay, leading to temporary cell cycle arrest to allow damage repair. By repressing the expression of the essential transcription termination factor *nusG*, OxyS enables read-through transcription into a cryptic prophage encoding *kilR*. The KilR protein interferes with the function of the major cell division protein FtsZ, thus imposing growth arrest. This transient growth inhibition facilitates DNA damage repair, enabling cellular recovery, thereby increasing viability following stress. The OxyS-mediated growth arrest represents a novel tier of defense, introducing a new regulatory concept into bacterial stress response.**

**Keywords** cell cycle arrest; checkpoint; *Escherichia coli*; prophage; small RNA
**Subject Categories** Cell Cycle; Microbiology, Virology & Host Pathogen Interaction; RNA Biology
**The EMBO Journal (2018) 37: 413–426**

## Introduction

The oxidative stress-induced OxyS small RNA (sRNA) was one of the first characterized sRNAs (Altuvia *et al*, 1997). A global regulatory role for OxyS was indicated by the substantial changes in the protein synthesis patterns observed with constitutive OxyS expression in *Escherichia coli* (Altuvia *et al*, 1997). Over the years, several approaches have been employed to identify genes controlled by OxyS including genetic screens, computational target prediction, and transcriptome analysis (Altuvia *et al*, 1997; Tjaden *et al*, 2006;

De Lay & Gottesman, 2012). OxyS was shown to negatively regulate the mRNAs of the transcription factors FhlA, RpoS, and FlhDC, as well as a number of additional proteins (Altuvia *et al*, 1997; Tjaden *et al*, 2006; De Lay & Gottesman, 2012). Whereas negative regulation of *rpoS* (stationary-phase sigma factor) appears to be indirect, that is, via titration of Hfq, regulation of *fhlA* and *flhDC* by OxyS results from direct base pairing with their mRNAs (Altuvia *et al*, 1998; Zhang *et al*, 1998; Argaman & Altuvia, 2000; Moon & Gottesman, 2011; De Lay & Gottesman, 2012).

Despite these findings, an intriguing aspect of OxyS remained enigmatic ever since its discovery; the RNA has been proposed to play a key role in protecting cells against the damaging effects of spontaneous and induced mutagenesis (Altuvia *et al*, 1997). However, attempts to reveal targets responsible for the sRNA antimutagenic phenotype failed. DNA can undergo various forms of damage by exposure to environmental stresses from various sources, or as a result of normal metabolism, producing genotoxic products. To prevent mutagenesis and maintain genome stability, DNA damage responses have evolved that control multiple processes including DNA repair, cell cycle checkpoints, and cell death (Spampinato, 2016). Here, we show that the OxyS antimutagenic phenotype is intricately linked with transient induction of cell growth arrest. By lowering the expression of the transcription termination factor NusG, OxyS enables expression of a *rac* prophage *kilR* gene encoding an inhibitor of cell division. We propose that the transient inhibition of cell division induced by OxyS facilitates DNA repair and recovery from oxidative stress.

NusG is a highly conserved protein regulator of RNA polymerase (RNAP). Through its N-terminal domain, it associates with RNAP modulating its processivity and termination properties (Sullivan & Gottesman, 1992; Werner, 2012). The C-terminal domain of NusG can either bind the transcription termination factor Rho (stimulating Rho-dependent termination) or NusE, a component of the 30S ribosomal subunit (S10) with dual roles in transcription and translation control. Consequently, under certain circumstances, the N-terminal domain of NusG is bound by RNAP, while the C-terminal domain associates with NusE. Simultaneous binding of NusG to RNAP and NusE directly links the elongating transcription complex to the

1    Department of Microbiology and Molecular Genetics, IMRIC, The Hebrew University-Hadassah Medical School, Jerusalem, Israel
2    Faculty of Biology, Genetics and Experimental Bioinformatics, University of Freiburg, Freiburg, Germany
3    Bioinformatics Group, Department of Computer Science, University of Freiburg, Freiburg, Germany
    *Corresponding author. Tel: +49 761 2032796; E-mail: wolfgang.hess@biologie.uni-freiburg.de
    **Corresponding author. Tel: +972 54 882 0623; E-mail: shoshy.altuvia@mail.huji.ac.il
    †These authors contributed equally to this work

ribosome and provides the physical framework for the coupling of transcription and translation in bacteria (Burmann *et al*, 2010).

Transcription termination is essential for cell viability and so are Rho and its cofactor, NusG (Downing *et al*, 1990). Studies of RNAP distribution and mapping of transcription elongation and termination in *E. coli* cells have provided an explanation for the essentiality of NusG. Rho activity clusters in horizontally acquired DNA fragments including prophages, indicating that Rho and NusG play an important role in silencing potentially harmful foreign genes (Cardinale *et al*, 2008; Mooney *et al*, 2009; Peters *et al*, 2009, 2012). Moreover, elimination of such foreign DNA elements permits deletion of *nusG*, although Rho itself remains essential (Cardinale *et al*, 2008). Here, we provide evidence that inhibition of cell division due to transient repression of *nusG* by OxyS protects cells from DNA damage.

## Results

### OxyS expression inhibits cellular growth

Studying OxyS, we noticed that plasmid-borne, unregulated expression of OxyS is detrimental. Moreover, we identified an *oxyS* point mutation (OxyS$_{A69C}$) that promoted toxicity beyond wild-type levels. Intrigued by this phenotype, we set to randomly mutagenize OxyS wild type and OxyS$_{A69C}$ and screened for OxyS suppressor mutants that suppressed toxicity. In light of our observation that OxyS toxicity was more pronounced in strains deficient for RelA, the stringent response major regulator, we used this genetic background to select for suppressor mutations. To eliminate mutations rendering OxyS completely inactive, for example, by decreasing sRNA stability, we focused on mutants that are no longer toxic, but capable of repressing *fhlA-lacZ*, a previously characterized target of OxyS (Argaman & Altuvia, 2000). We isolated and sequenced 25 mutants including some carrying identical mutations. Mutations that rendered OxyS harmless clustered in two sites: the loop sequence of hairpin B (OxyS$_{C56U; C58U}$) and the single-stranded region located between hairpins B and C (OxyS$_{C76U; C77U}$ and OxyS$_{C70U}$) (Fig 1A). The nontoxic mutant OxyS$_{C76G; C77G}$ was constructed by site-directed mutagenesis based on the screen above. For P*lac*-controlled expression, the mutations were transferred to pBR-P*lac* plasmid (Guillier & Gottesman, 2006), and the mutants were examined for their effect on cellular growth upon induction with IPTG. Growth curves and survival assays showed that whereas wild-type OxyS and OxyS$_{A69C}$ inhibited growth, forming only few CFU, the growth arrest phenotype was no longer detectable in the OxyS$_{C56U; C58U}$ and OxyS$_{C76U; C77U}$ mutants (Fig 1B and Appendix Fig S1A). The revertant of OxyS$_{A69C}$, OxyS$_{A69C; C70U}$, carrying both mutations, exhibited an intermediate growth rescue. Northern blots showed that the RNA levels of the suppressor OxyS mutants were comparable to wild-type OxyS (Fig 2B), and functional assays showed that OxyS mutants repressed the translation of *fhlA-lacZ* (Appendix Fig S1B), indicating that the OxyS mutants were active regulators, though not toxic.

### OxyS represses the *nusG* mRNA

The availability of highly toxic and nontoxic OxyS mutants prompted us to search for targets whose putative complementary sites match the observed changes in OxyS mutants. We performed a whole genome IntaRNA (Wright *et al*, 2014) search in *E. coli* with wild-type OxyS and its suppressor mutant versions. Of 4,301 investigated genes, 66 matched the predicted pattern. To further pin down the critical target(s), this list of genes was compared with the list of *E. coli* essential genes (Baba *et al*, 2006), yielding three candidates (*nusG*, *pyrG*, and *orn*). Here, we present the physiological consequences of OxyS-*nusG* interaction.

The *nusG* gene is second in the operon *secE-nusG,* and OxyS is predicted to base-pair with the ribosome-binding site of *nusG* (Fig 2A). The mutation OxyS$_{A69C}$ is expected to extend the interaction with the Shine–Dalgarno sequence and to increase the stability of *oxyS-nusG* hybrid. Northern Blot analysis detected a 2.5-fold to 3.5-fold reduction in *nusG* mRNA levels upon exposure to wild-type OxyS or OxyS$_{A69C}$ pulse expression compared to OxyS suppressor mutants, indicating that expression of *nusG* is negatively regulated by OxyS (Fig 2B). To examine the effect of OxyS on *nusG* at the post-transcriptional level, we constructed a P*tac-nusG-lacZ* translational fusion. Levels of *nusG-lacZ* were reduced twofold to threefold in the presence of OxyS or OxyS$_{A69C}$, whereas the isolated OxyS suppressor mutants had no effect on *nusG-lacZ* expression (Fig 2C). Given that stable RNA hybrids may block the elongation by reverse transcriptase, we examined the interaction of *in vitro*-synthesized OxyS and *nusG* RNAs using primer extension assays. These assays showed that the interaction between OxyS and *nusG* results in a strong termination signal (Fig 2D). The termination site mapped to the Shine–Dalgarno sequence of *nusG* that is the center of complementarity between OxyS and *nusG*. The termination signal produced by OxyS$_{A69C}$ was stronger than that of wild type, while the suppressor mutants exhibited a weaker signal. No termination was detected using OxyS$_{C76G; C77G}$. An RNase protection assay further confirmed these findings. In this assay, base-pairing RNAs are protected while unpaired nucleotides are cleaved by single-stranded specific ribonucleases. As the binding between OxyS and *nusG* is discontinuous, base-pairing efficiency between *nusG* and OxyS mutant RNAs was estimated based on the formation of the full-length hybrid (Appendix Fig S2A). The levels of the full-length hybrid formed with OxyS$_{A69C}$ were higher than those formed with wild-type OxyS, whereas OxyS$_{C76G; C77G}$ showed decreased levels of hybrid protection.

To affirm base pairing, we replaced the GG nucleotides at positions −15 and −16 in *nusG* mRNA by CC. This change was predicted to complement OxyS$_{C76G; C77G}$. The P*tac-nusG*$_{G-15C; G-16C}$-*lacZ* translational fusion was slightly repressed by wild-type OxyS; however OxyS$_{C76G; C77G}$ failed to restore repression by binding its complementary mutant *nusG*$_{G-15C; G-16C}$, indicating that repression may involve more than one interaction site (Fig 3A and B, left panel). Searching the sequence nearby, we identified another site for OxyS binding slightly upstream of the site overlapping the RBS (Fig 3A). This site of *nusG* was predicted to bind a sequence at the 3′ end of OxyS from nucleotide G75 to U87, suggesting that *nusG* mRNA can be bound by two different OxyS molecules. The P*tac-nusG*$_{G-31C; G-32C}$-*lacZ* translational fusion was repressed by wild-type OxyS, whereas repression by its complementary OxyS mutant OxyS$_{C76G; C77G}$ was less prominent (Fig 3B). Expression of NusG quadruple mutant (G-15C; G-16C and G-31C; G-32C) carrying mutations at the two sites predicted to bind the same sequence in OxyS was repressed by OxyS$_{C76G; C77G}$ and unaffected by wild-type OxyS (Fig 3B right panel). Together, the data indicate that two molecules of OxyS can simultaneously bind two different sites in *nusG*.

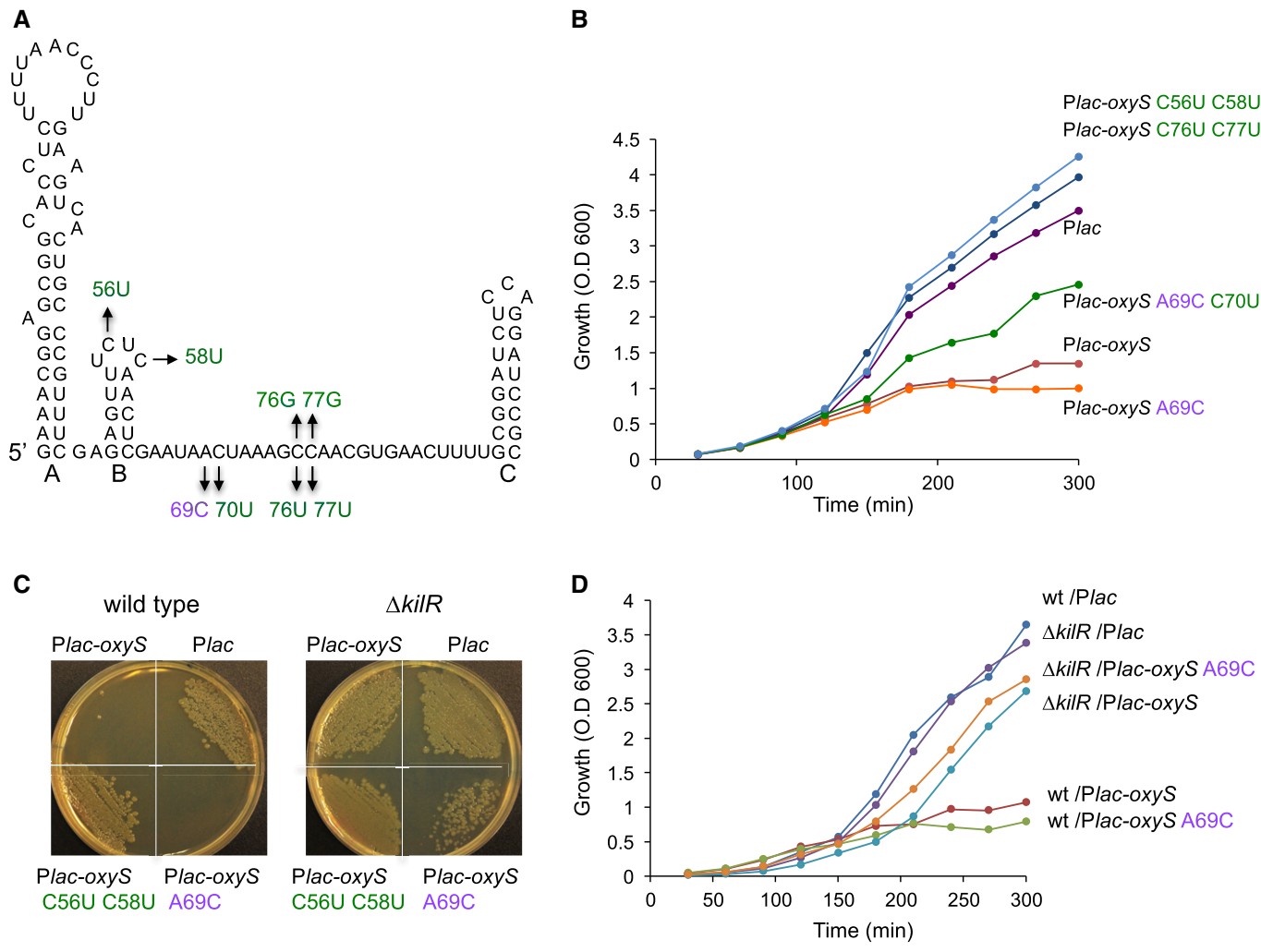

**Figure 1. Toxic and nontoxic OxyS.**

A   Toxic (purple) and nontoxic suppressor mutations (green) in OxyS. The loops of hairpins A and C were found previously to interact with the *fhlA* ribosome-binding site. Hairpin C is Rho-independent transcription terminator.

B   Growth curves of cells (MG1655 *relA::cat*, *lacI*q) with OxyS plasmids. Cultures carrying plasmids were treated with 1 mM IPTG at dilution, OD was measured as indicated.

C   OxyS is not toxic in *kilR*-deficient cells. Wild-type and Δ*kilR::cat* cells were transformed with plasmids expressing OxyS, wild type and mutants, as indicated.

D   Growth curves of wild-type and *kilR* mutant cells with OxyS plasmids. Cultures of wild-type and Δ*kilR::cat* carrying plasmids were treated with 1 mM IPTG at dilution, OD was measured as indicated. Both strains are also *relA::kan*, *lacI*q. Although the plating of OxyS$_{A69C}$ in Δ*kilR* seems less efficient than that of OxyS, their growth curves are very similar, indicating that neither OxyS nor OxyS$_{A69C}$ are toxic in Δ*kilR*.

Analysis of RNA interactions by EMSA further showed that *nusG* quadruple mutant could be bound by its complementary OxyS$_{C76G;}$ $_{C77G}$ mutant as opposed to wild-type OxyS (Appendix Fig S2B).

**The toxic phenotype of OxyS results from *nusG* repression**

NusG has been implicated in the suppression of toxic activities by horizontally acquired genes (Cardinale *et al*, 2008). Furthermore, deletion of the cryptic *rac* prophage in wild-type *E. coli* permits deletion of *nusG*. The *rac* prophage carries the *kilR* gene encoding an inhibitor of cell division (Conter *et al*, 1996; Burke *et al*, 2013). To determine whether the toxic phenotype of OxyS is due to activation of *kilR*, we transformed plasmid P*lac-oxyS* into an *E. coli* strain lacking

the *kilR* gene. Wild-type *E. coli* failed to yield any colonies when transformed with P*lac-oxyS* or P*lac-oxyS*$_{A69C,}$ whereas mutation of *kilR* supported colony formation (Fig 1C). The suppressor nontoxic OxyS mutant formed normal size colonies in both wild type and Δ*kilR*. Similarly, growth of wild-type cells expressing OxyS and OxyS$_{A69C}$ was inhibited, whereas Δ*kilR::cat* cells remained unaffected (Fig 1D). Together, these data indicate that toxicity is due to elevated expression of KilR caused by OxyS-mediated repression of *nusG*.

Furthermore, as *nusG* is second in the operon *secE-nusG*, it is possible that OxyS also decreases *secE* expression. However, such a decrease is not relevant for the toxic phenotype of OxyS. OxyS is less toxic in the presence of a plasmid expressing *nusG*, whereas in the presence of a plasmid expressing *secE*, OxyS is highly toxic

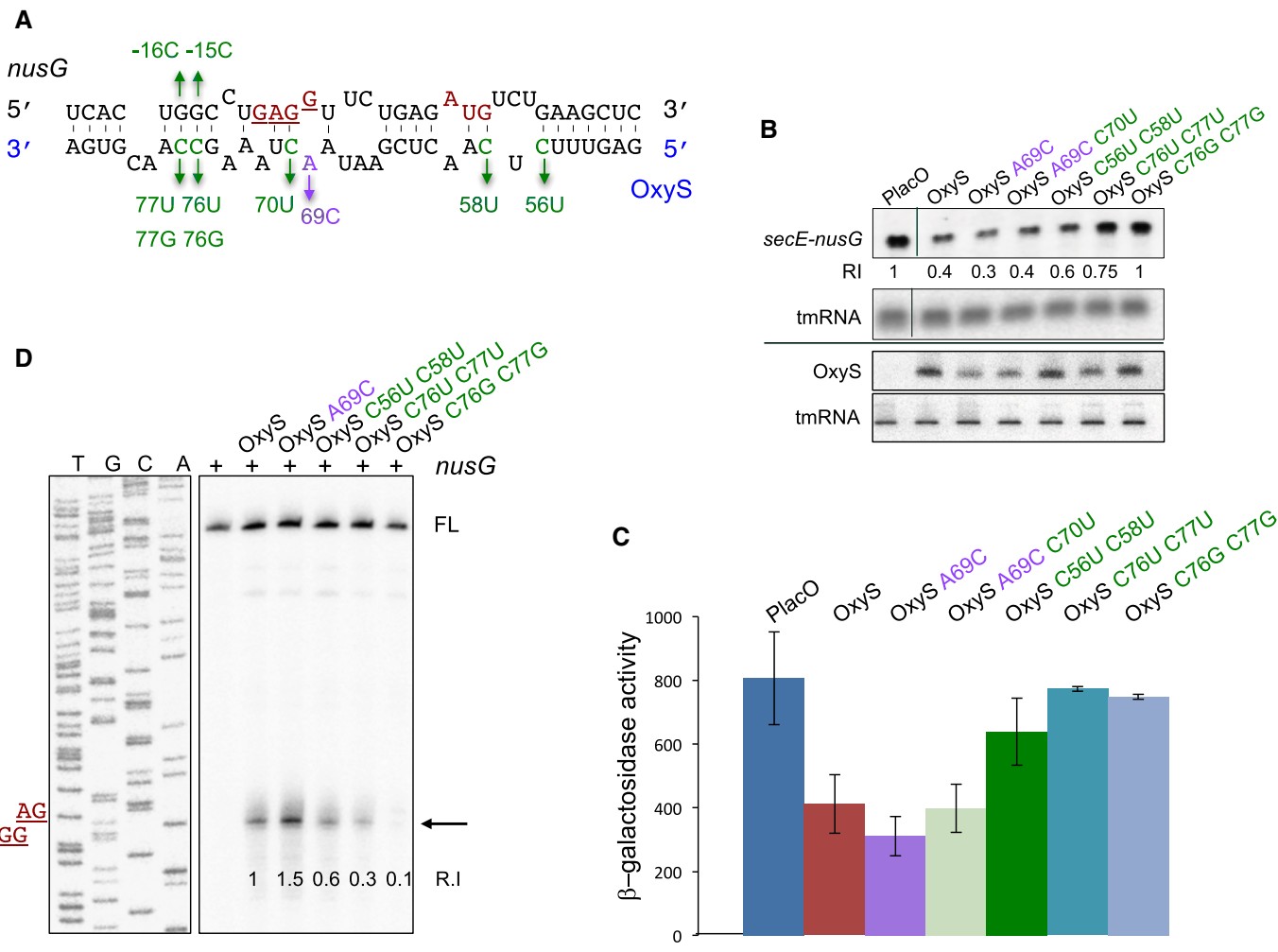

**Figure 2. OxyS represses *nusG* expression.**

A   Predicted base pairing between *nusG* and OxyS RNAs. The initiation codon and the Shine–Dalgarno sequence of *nusG* are marked in red. Suppressor nontoxic OxyS mutants are in green, the highly toxic mutant is in purple.

B   Northern analysis of RNA extracted from cells carrying control and OxyS plasmids. Total RNA was extracted from cultures treated at dilution with 1 mM IPTG for 90 min. Samples were analyzed by 1.4% agarose–formaldehyde gel electrophoresis to detect the full-length transcript of *secE-nusG* and tmRNA or 6% urea–PAGE to detect OxyS and tmRNA. The membranes were probed with 5′ end-labeled OxyS and tmRNA-specific primers and antisense riboprobe to detect *secE-nusG* mRNA. Relative intensity (RI) of *secE-nusG* in the presence of control, wild-type, and OxyS plasmids.

C   Cultures (*relA::cat, ΔoxySli::frt, lacZ::Tn10, lacI^q*) carrying P*tac-nusG-lacZ* (pSC101*; single copy) translational fusion and OxyS plasmids were treated with IPTG (1 mM) at OD$_{600}$ = 0.1. β-Galactosidase activity was measured 120 min after treatment. Results are displayed as mean of two to eight biological experiments ± standard deviation.

D   Primer extension of *in vitro*-synthesized *nusG* mRNA in the absence or presence of synthesized OxyS RNAs. Full-length cDNA (FL). Arrow denotes termination signal. The site of Shine–Dalgarno sequence (GGAG) is denoted in red. Relative intensity (R.I) denotes the ratio of the termination signal per full-length compared to wild-type *oxyS-nusG* interaction, which was used as a 100% reference.

Source data are available online for this figure.

(Appendix Fig S3). These results indicate that an increase in the expression levels of *nusG* can negate *oxyS* toxicity, whereas a concomitant increase in *secE* has no effect on *oxyS* toxicity. Therefore, *oxyS* is toxic because it decreases *nusG* expression levels.

## The toxic phenotype of OxyS correlates with bacterial recovery from stress

Expression of OxyS is induced upon exposure to hydrogen peroxide (Altuvia *et al*, 1997). To determine whether OxyS expression

influences recovery of *E. coli* exposed to oxidative stress, and to investigate whether OxyS toxicity is required for efficient recovery, we transformed P*lac-oxyS* (toxic) and P*lac-oxyS*$_{C56U; C58U}$ (nontoxic) into an *oxyS* mutant in which only the regulatory elements relevant for OxyS toxicity were deleted. In this mutant, the overlapping *oxyR-oxyS* promoters, the 5′ end hairpin of OxyS, and its 3′ end *rho*-independent terminator were left intact to avoid polar effects (Δ*oxySli*). Cultures carrying control and OxyS expressing plasmids were grown with IPTG to induce OxyS expression in the exponential phase, when the cultures were treated with hydrogen peroxide to

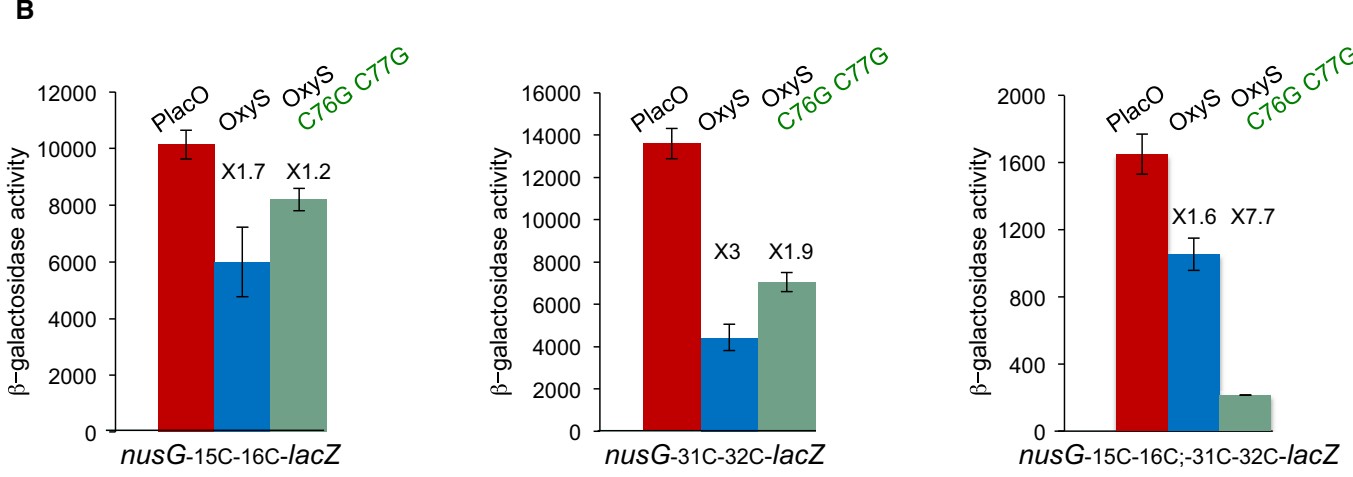

**Figure 3. Two OxyS molecules can bind *nusG* at two different sites.**

A  Extended base pairing between *nusG* and OxyS RNAs. The initiation codon and the Shine–Dalgarno sequence of *nusG* are marked in red. Suppressor nontoxic OxyS mutants are in green, the highly toxic mutant is in purple. The OxyS two molecules are marked in blue. *nusG* quadruple mutant (G-15C; G-16C; G-31C; G-32C).

B  Cultures carrying P*tac*-*nusG*-*lacZ* (double and quadruple mutants) and OxyS (wild type and C76G; C77G) were treated with IPTG (1 mM) at OD$_{600}$ = 0.1. β-Galactosidase activity was measured 120 min after treatment. Results are displayed as mean of three biological experiments ± standard deviation. Fold repression of *nusG* mutants by wild type and OxyS mutant is denoted. The changes in the basal expression levels of *nusG* mutants could be due to an effect of the mutations on either the sequence or the structure encompassing the RBS.

induce oxidative stress. After 30 min of treatment, the cultures were washed and resuspended in fresh medium. Recovery was determined after 60 min of growth in nonstressful medium (after wash). Although the survival of cultures expressing wild-type OxyS decreased dramatically (Appendix Fig S1A), it was higher than cultures carrying a control plasmid after exposure to hydrogen peroxide. The recovery rate of the nontoxic OxyS mutant was between the rate of the vector control and OxyS, demonstrating that OxyS toxicity has an important role in bacterial recovery (Fig 4A).

The levels of OxyS at 30 min after exposure to hydrogen peroxide are about sevenfold lower than OxyS levels produced from the P*lac* plasmid (Appendix Fig S4). To investigate the effect of the chromosomally encoded *oxyS* allele on recovery from oxidative stress, and to determine the role of *kilR* in *oxyS*-mediated recovery, cultures of wild-type Δ*oxyS* and Δ*kilR* exposed to hydrogen peroxide for 30 min were examined at 60 min after H$_2$O$_2$ removal. Survival of cells carrying an intact chromosomally located *oxyS* allele increased by ~4-fold, whereas survival of Δ*oxyS* and Δ*kilR* mutants decreased by ~1.5-fold (Fig 4B), indicating that *oxyS*-mediated recovery required *kilR*. Furthermore, quantification of *kilR* RNA level using real-time PCR demonstrated that upon exposure to H$_2$O$_2$, *kilR* mRNA levels increased by ~3-fold in wild-type cells carrying an intact *oxyS*

allele compared to *oxyS*-deficient cells (Fig 4C). The increase in *kilR* levels followed a decrease in *nusG* mRNA stability; the half-life of *nusG* mRNA in wild-type cells exposed to hydrogen peroxide was reduced by 1.5-fold compared to its half-life in *oxyS* mutant (Fig 4D and E). Likewise, NusG:SPA protein levels carrying the sequential peptide affinity (SPA) tag at the NusG C-terminal end decreased upon exposure to H$_2$O$_2$ in wild type but not in an *oxyS*-deficient mutant (Fig 4F). Together, the results demonstrate that *kilR* expression was controlled by *nusG* and that expression of the latter was inhibited by OxyS.

## OxyS promotes recovery from stress by interfering with cell division

Assembly of the essential, tubulin-like FtsZ protein into a ring-shaped structure at the nascent division site serves as a scaffold for recruitment of the cell division machinery. Kil proteins prevent cell division by interfering with FtsZ function (Conter *et al*, 1996; Burke *et al*, 2013; Haeusser *et al*, 2014; Hernandez-Rocamora *et al*, 2015). To determine whether recovery is due to interference with cell division, we examined growth resumption in cells with an intact *oxyS* allele and mild overexpression of *ftsQAZ* operon encoding *ftsQ*, *ftsA*,

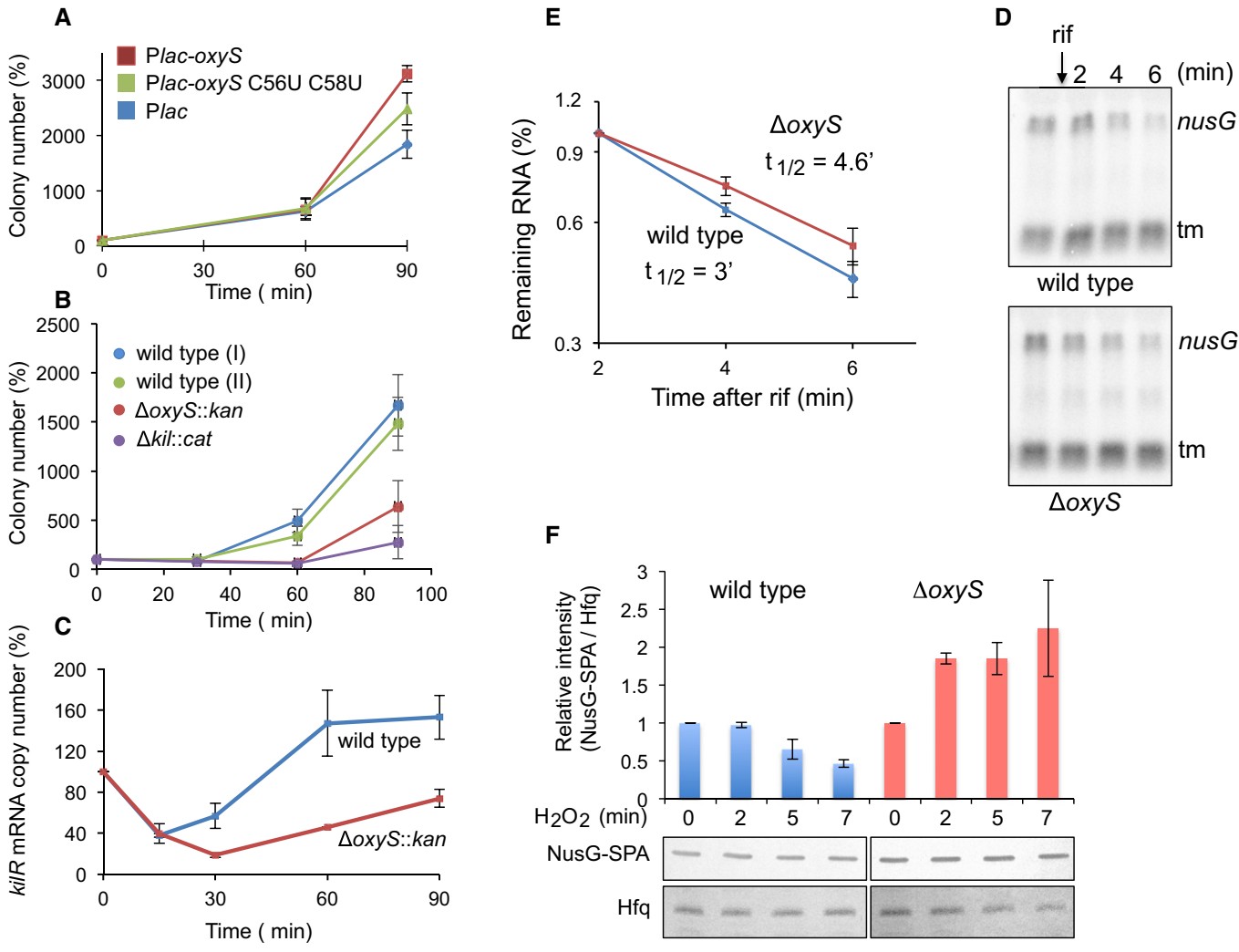

**Figure 4. OxyS facilitates recovery from oxidative stress.**

A   OxyS toxicity facilitates recovery. Cultures (*relA::cat ΔoxySli::kan, lacI*$^q$) with OxyS plasmids grown with IPTG (1 mM) were treated with 1 mM H$_2$O$_2$ at OD$_{600}$ = 0.1 for 30 min. Thereafter, the cultures were washed and continued to grow in fresh LB medium. Samples were taken 30, 60, and 90 min after wash. The number of cells after wash was used as 100% reference. Results are displayed as mean of two biological experiments ± standard deviation.

B   OxyS-mediated recovery requires the function of KilR protein. Cultures as indicated were treated with H$_2$O$_2$ for 30 min and washed as described above. *relA::cat* (wild-type I) and *relA::kan* (wild-type II) were used as controls for *ΔoxySli::kan* and *ΔkilR::cat*, respectively. Results are displayed as mean of four biological experiments ± standard deviation.

C   OxyS increases *kilR* mRNA levels in response to oxidative stress. RT–PCR of RNA samples taken at the indicated time points following exposure to 1 mM of H$_2$O$_2$. Two samples per treatment and two reactions per sample were analyzed. Results are displayed as mean of two biological experiments ± standard deviation. *kilR* initial levels detected in the absence of treatment were used as 100% reference.

D   Northern analysis of *secE-nusG* mRNA in wild type and *oxyS* mutant exposed to hydrogen peroxide prior to the addition of rifampicin (rif).

E   Calculated half-life of *nusG* mRNA in wild type and *oxyS* mutant following exposure to H$_2$O$_2$. Average and standard deviations of two biological experiments are shown.

F   OxyS decreases NusG-SPA protein levels in response to oxidative stress. Wild-type and *oxyS* mutant cells carrying NusG-SPA were exposed to 1 mM H$_2$O$_2$. Protein samples taken at the indicated time points were analyzed using SPA-specific antibodies. The intensities of NusG-SPA and Hfq (serving as a loading control) were measured using Image Studio Lite program. Relative intensity was calculated using NusG-SPA initial levels (in the absence of treatment) as 100% reference. Standard deviations of two biological experiments are shown.

Source data are available online for this figure.

and *ftsZ* genes from a low-copy plasmid (Bernhardt & de Boer, 2004). The presence of *ftsQAZ* abrogated bacterial recovery from stress (Appendix Fig S5A). Together, the data indicate that OxyS-mediated cell division inhibition is crucial for bacterial growth resumption following stress.

We visualized the effect of OxyS on cell morphology by fluorescence microscopy. Images of cells expressing OxyS or OxyS$_{A69C}$ displayed elongated filaments due to lack of cell division. In contrast, cells expressing the nontoxic mutant OxyS$_{C56U; C58U}$ formed short rod-shaped cells very similar to control cells (Fig 5A).

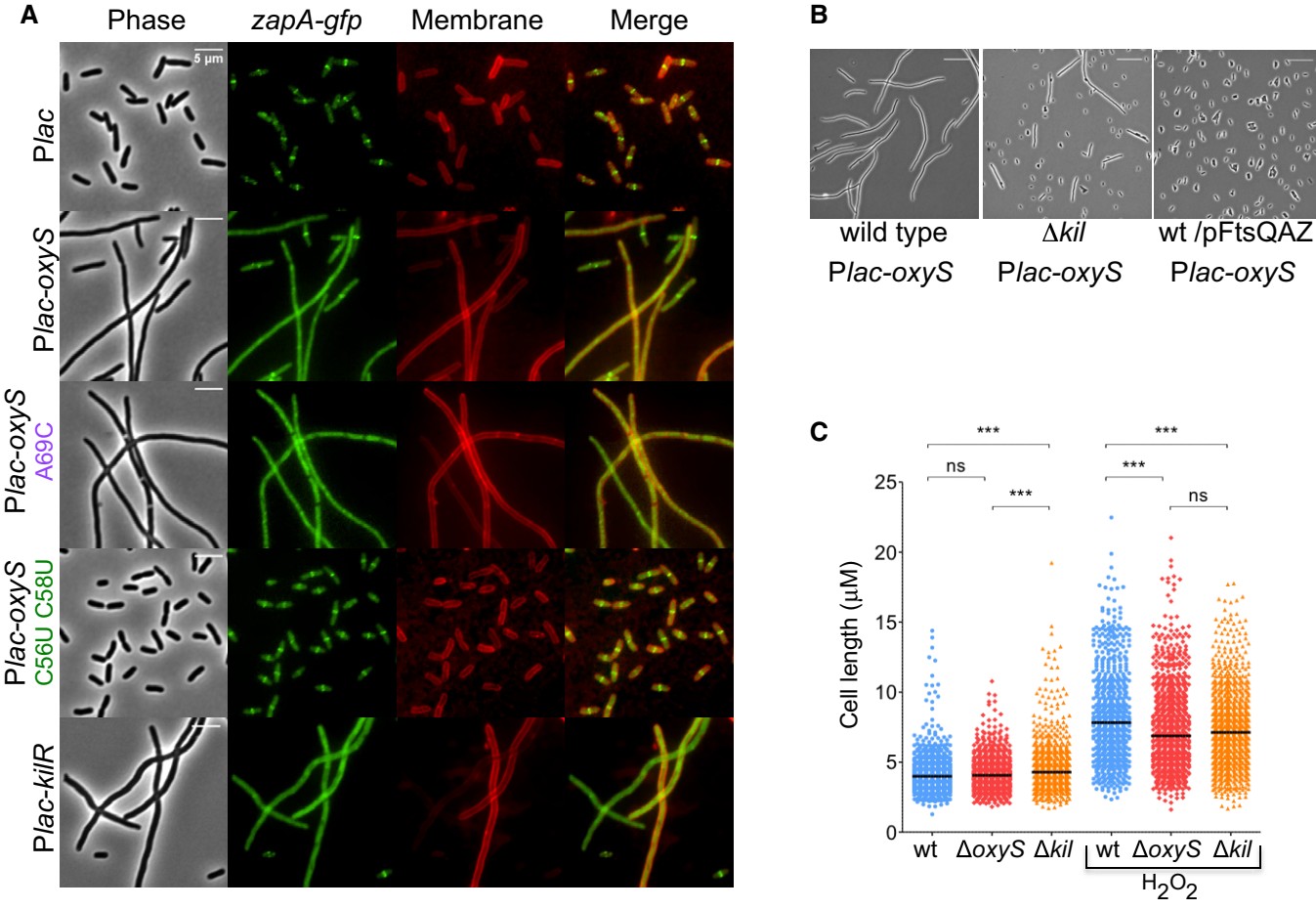

**Figure 5.  Fluorescence microscopy images.**

A   OxyS expression impairs cell division. Cultures of *Escherichia coli* carrying *ZapA-gfp* fusion as a single copy in the native position in the chromosome (*ZapA-gfp:cat*) and P*lac-oxyS* plasmids were treated with 1 mM IPTG at dilution. Samples were taken at 3 h post-dilution. All samples were spotted on PBS agar pad for imaging. DNA stained blue with DAPI. Scale bar, 5 µm.

B   OxyS-mediated impaired cell division is prevented in Δ*kilR* mutant and by overexpression of FtsQAZ. Cells were grown as described above for 3 h at 37°C in the presence of 1 mM IPTG. The operon FtsQAZ is expressed from its own promoter. Scale bar, 5 µm.

C   OxyS induced by $H_2O_2$ impairs cell division. Scatter plots of cell length distribution of wild-type Δ*oxyS* and Δ*kilR* grown without or with $H_2O_2$ treatment. The cultures at $OD_{600}$ = 0.1 were exposed to 1 mM $H_2O_2$ or remained untreated. Cell lengths were measured 60 min thereafter. The black line in each plot represents the median of three biological experiments. In each experiment, more than 750 cells were analyzed (GraphPad Prism software; unpaired *t*-test, ***P*-value = 0.0001).

The ZapA protein interacts with FtsZ at an early stage of FtsZ-ring assembly and co-localizes with it (Gueiros-Filho & Losick, 2002). Images of cells with *zapA-gfp* fusions showed that ring assembly was impaired in cells expressing OxyS (Fig 5A). The OxyS-mediated filamentation was prevented in the *kilR* mutant and by overexpression of the *ftsQAZ* operon from a low-copy plasmid (Fig 5B and Appendix Fig S5B). Statistical analysis of cell length distribution of cultures expressing OxyS in wild type or Δ*kilR* and of wild-type cultures expressing both OxyS and FtsQAZ further confirmed the conclusion that OxyS-mediated filamentation requires active KilR and inactive FtsZ (Appendix Fig S5B). Moreover, images of cells expressing a plasmid-encoded KilR protein displayed elongated filaments and impaired ring assembly (Fig 5A).

Significantly, we find that the presence of an intact *oxyS* allele affects cell length in response to oxidative stress. The length of wild-type Δ*oxyS* and Δ*kilR* cells is comparable prior to exposure to $H_2O_2$

(Fig 5C), increasing after treatment. Importantly, cells expressing OxyS are longer than Δ*oxyS* or Δ*kilR* mutants, indicating that in response to oxidative stress, expression of chromosomally encoded *oxyS* impairs cell division.

## The antimutator phenotype of OxyS is due to *nusG* repression by OxyS

In our previous study, we showed that OxyS protects cells against DNA damage, decreasing the rate of mutations of spontaneous or induced mutagenesis (Altuvia *et al*, 1997). To determine whether the reduced rate of mutations is due to the regulatory cascade that begins with OxyS-mediated repression of *nusG*, we examined the number of rifampicin-resistant mutants following exposure to hydrogen peroxide in Δ*oxyS* cells carrying plasmids expressing OxyS or $OxyS_{C56U;\ C58U}$. The number of rifampicin-resistant mutants

**Table 1. OxyS protects against DNA damage**

| Rif$^r$ mutants induced by 5 mM $H_2O_2$ | |
|---|---|
| Genotype | Rif$^r$ mutants per $10^8$ cells |
| Δ*oxySli::frt/Plac* | 68.3 ± 36.0 |
| Δ*oxySli::frt/Plac-oxyS* | 2.2 ± 1.8 |
| Δ*oxySli::frt/Plac-oxyS*$_{C56U;C58U}$ | 37.4 ± 20.0 |
| In Δ*kilR* mutant | Rif$^r$ mutants per $10^8$ cells |
| Δ*oxySli::frt, ΔkilR::cat/Plac* | 41.3 ± 1.2 |
| Δ*oxySli::frt, ΔkilR::cat/Plac-oxyS* | 62.3 ± 5.1 |
| Δ*oxySli::frt, ΔkilR::cat/Plac-kilR* | 2.0 ± 1.0 |
| In the presence of pFtsQAZ | Rif$^r$ mutants per $10^6$ cells |
| Δ*oxySli::frt/pFtsQAZ, Plac* | 65.3 ± 2.4 |
| Δ*oxySli::frt/pFtsQAZ, Plac-oxyS* | 493.7 ± 229.3 |
| Rif$^r$ mutants induced by 1 mM $H_2O_2$ | |
| Chromosomal *oxyS* | Rif$^r$ mutants per $10^8$ cells |
| Wild type | 12.3 ± 3.3 |
| Δ*oxyS* | 68.6 ± 8.0 |

Results are displayed as mean of four to five biological experiments ± standard deviation.

normalized to the number of viable cells demonstrated that OxyS toxicity correlated with a reduced number of mutants; the nontoxic OxyS suppressor mutant exhibits a higher fraction of Rif$^r$ mutants, similar to the numbers determined with the control cells (Table 1). The reduction in mutation rates was no longer detected in the Δ*kilR* mutant, or when FtsQAZ was overexpressed, further confirming that OxyS acts an antimutator by interfering with cell division. Moreover, cells with in *trans* expression of *rac* KilR from a plasmid exhibited a low number of Rif$^r$ mutants similar to cells with OxyS expression (Table 1). This indicates that NusG signaling can be bypassed by expressing KilR, the last component in the *oxyS-nusG-kilR* cascade.

Importantly, we demonstrate that hydrogen peroxide-dependent induction of OxyS results in a dramatic reduction in the number of rifampicin-resistant mutants (Table 1).

## Discussion

Our previous study demonstrated that OxyS protects cells from DNA damage (Altuvia *et al*, 1997). Here, we show that OxyS is toxic and that the protection from DNA damage is intricately linked with OxyS toxicity. Acting as a base-pairing RNA, OxyS decreases the expression of the essential transcription termination factor NusG. This decrease in transcription termination leads to an increase in the expression of the prophage-encoded *kilR* gene. The KilR protein interferes with FtsZ function and thus inhibits cell division. The OxyS-induced regulatory cascade results in bacterial growth arrest, thus facilitating DNA damage repair following stress.

To protect genomic integrity and to promote survival of cells exposed to genotoxic agents, prokaryotic and eukaryotic cells activate DNA damage response (DDR) pathways. In eukaryotes, the DDR is a signal transduction pathway that upon sensing of DNA

damages promotes multiple physiological processes including apoptosis, senescence, activation of immune surveillance, and DNA repair. Many of these responses depend on post-translational modifications such as phosphorylation. Some are controlled by slowed transcription. An intriguing example is p53, the potent tumor suppressor protein and core transducer of the DDR. Exposure to DNA damage leads to rapid activation of p53, which in turn induces cell cycle arrest, apoptosis, or senescence. The activation of p53 in response to damage is transient and thus provides the cell with a mechanism that connects DNA repair with cell cycle progression (Ciccia & Elledge, 2010). Here, we establish a link between two enigmatic OxyS phenotypes: the toxic phenotype and the antimutagenic phenotype. We show that a cellular programming mediated by OxyS toxicity is at the heart of its antimutagenic phenotype; OxyS is toxic because it interferes with FtsZ function; however, by inhibiting cell division, OxyS enables the bacterial repair systems to repair damage for a longer period of time, thus leading to an increase in the rate of recovery. The observed reduction in DNA damage and greater recovery are both mediated by the *rac* prophage-encoded *kilR* gene.

The *E. coli* genome contains nine cryptic prophage elements. Their expression provides protection from sublethal concentrations of quinolone and β-lactam antibiotics, primarily through small proteins that inhibit cell division such as *kilR* of Rac prophage and *dicB* of Qin prophage. Furthermore, these prophages help bacteria to cope with adverse environmental conditions providing increased resistance to osmotic, oxidative, and acid stress (Wang *et al*, 2010). Although λ *kil* and *rac kilR* share no sequence similarities, their function is similar; λ *kil* prevents proper FtsZ assembly, producing shorter oligomers by disrupting FtsZ protofilaments while sequestering the protein subunits (Haeusser *et al*, 2014; Hernandez-Rocamora *et al*, 2015). Consequently, λ *kil* expression results in cell division inhibition and filamentation. Likewise, although no details on the mechanism have been reported, *kilR* of *rac* phage prevents *E. coli* from dividing, causing cells to grow into long filaments that become nonviable. *kilR*-mediated inhibition of cell division is relieved by excess FtsZ (Conter *et al*, 1996; Burke *et al*, 2013). Cell division is also inhibited following DNA damage, as part of the SOS response. Cell survival requires coordination of division with other processes, and SulA of the SOS system inhibits cell division until genetic errors are corrected (Huisman *et al*, 1984; Bi & Lutkenhaus, 1993; Chen *et al*, 2012). SOS-mediated inhibition of cell division involves sequestration of FtsZ subunits similar to that described for *kil* expression. This mechanism is different from that employed by division site selection antagonists, such as SlmA that binds specific regions on the chromosome preventing FtsZ from assembling over unsegregated nucleoids, or MinC that prevents assembly of FtsZ in DNA-free regions of cell poles (Haeusser *et al*, 2014). In any case, as OxyS is toxic in SOS off-cells, the effect of OxyS is independent of SOS response involving SulA (Appendix Fig S6).

Here, we revealed an OxyS-induced molecular checkpoint relay, leading to temporary cell cycle arrest to allow damage repair. We propose that in addition to a first tier of defense that is aimed at elimination of reactive oxygen species and repair of cellular damages, *E. coli* employs a second tier of defense targeting cell division by modulating NusG levels. The transient growth inhibition caused by interference with cell division enables the bacterial repair systems to repair damage for a longer period of time, thus allowing

a higher number of cells to resume growth. We show that by decreasing *nusG* expression, the oxidative stress-induced OxyS sRNA leads to an increase in KilR and subsequently to inhibition of cell division (Fig 6). As the production of OxyS in response to oxidative stress is transient, decreasing dramatically by 60 min of exposure, and the levels of *kilR* reach a plateau at about the same time, the *oxyS*/*nusG*/*kilR*-dependent regulatory cascade is transient, leading to temporary cell cycle arrest.

The presence of an intact RelA allele seems to have an effect on the toxic phenotype of OxyS. In the *relA* mutant strain, both wild type and OxyS$_{A69C}$ exhibit a similar growth arrest phenotype, whereas in wild-type cells carrying an intact RelA allele, toxicity of OxyS$_{A69C}$ is more pronounced than that of wild-type OxyS (Appendix Fig S7). The difference in toxicity of OxyS mutants could be attributed to the observation that RelA-mediated synthesis of ppGpp in LB medium increases FtsZ protein levels, possibly through transcriptional regulation (Powell & Court, 1998). The increase in FtsZ protein levels could possibly compensate for the interference with FtsZ function.

Intriguingly, our experiments indicate that two molecules of OxyS can bind one *nusG* molecule at the same time. The first binding site spans nucleotides −40 to −29 with respect to the first nucleotide of the start codon. The second experimentally verified interaction site in *nusG* spans from nucleotide −20 to +13 with respect to the AUG. Given that OxyS$_{C76G; C77G}$ fails to repress its complementary mutant *nusG*$_{G-15C; G-16C}$, it suggests that the upstream site aids OxyS in binding of the downstream site overlapping the ribosome-binding region. Expression of NusG quadruple mutant (G-15C; G-16C and G-31C; G-32C) carrying mutations at the two sites predicted to bind the same sequence in OxyS was repressed by OxyS$_{C76G; C77G}$ and unaffected by wild-type OxyS. Together, the data indicate that two molecules of OxyS can

simultaneously bind two different sites in *nusG*. sRNAs interacting with multiple sites within an mRNA have been described before. In most cases, pairing with either site was found to be sufficient for full regulation. For example, the RybB sRNA can base-pair with either one of two mutually exclusive pairing sites within the translated portion of OmpD mRNA. However, pairing with either site is sufficient for OmpD regulation (Balbontin *et al*, 2010). SgrS sRNA binds two different sites in *manXYZ* operon, each capable of translation repression of different genes of the operon, although both sites are needed for RNase E-dependent degradation of the full-length transcript (Rice *et al*, 2012). The long *lrp* leader is bound by GcvB at two independent sites (Lee & Gottesman, 2016). Pairing with either site was sufficient for significant repression of *lrp* expression, although pairing at both sites gave the best repression (Lee & Gottesman, 2016). Spot 42 interacts with a number of targets via two different sites. Both sites were shown to contribute to regulation; however, whether Spot 42 base pairs with both sites simultaneously is unclear (Beisel *et al*, 2012). Further biochemical analyses will be required to fully understand the OxyS dual binding of *nusG* and the involvement of Hfq in this unique base-pairing regulatory mechanism in which two molecules of OxyS repress one molecule of *nusG* mRNA.

The small RNA OxyS is highly conserved. An OxyS homology search based on iterative blasting and structural filtering implemented in the GLASSgo web server (http://rna.informatik.uni-freiburg.de/GLASSgo/Input.jsp) detected 603 potential OxyS homologs, which are widely distributed in the enterobacterial clade as they belong to 10 different genera representing at least 27 different species and more than 500 different strains (Appendix Fig S8). All homologs are located in the 5′ intergenic region of the *oxyR* gene, sharing a widely conserved gene synteny (Appendix Fig S9). Multiple alignments of 36 randomly selected OxyS homologs exhibited

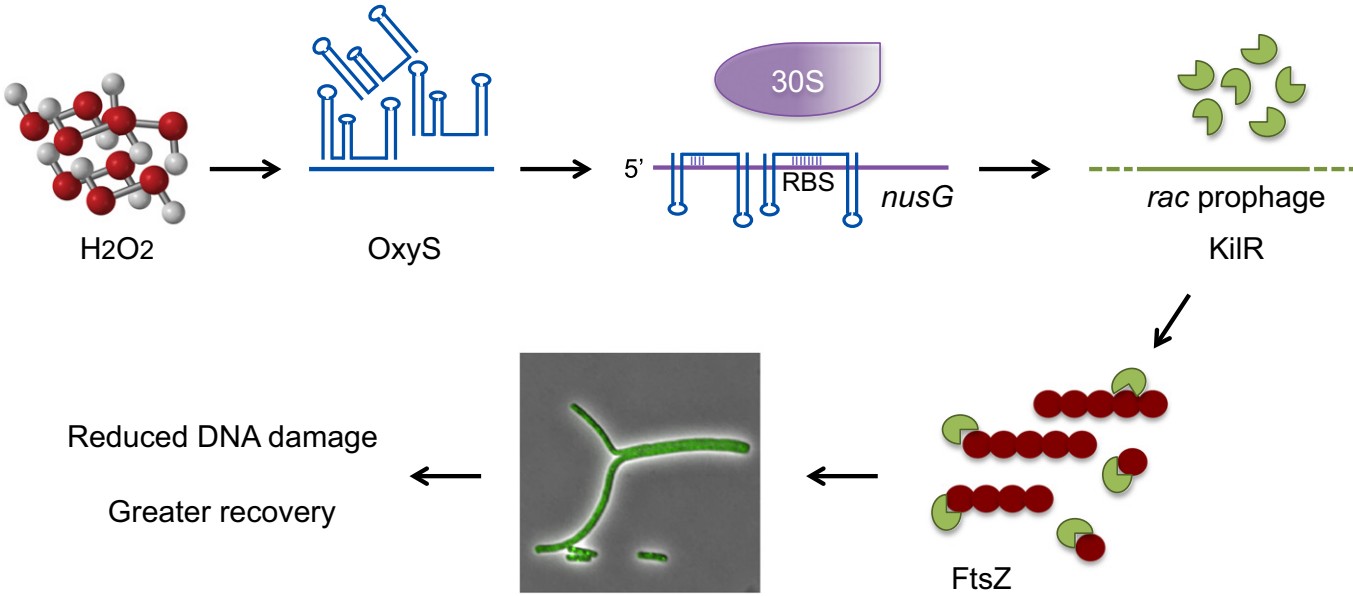

**Figure 6.  The transient growth arrest caused by cell division inhibition allows a greater number of cells to resume growth.**

By decreasing *nusG* expression, the oxidative stress-induced OxyS sRNA leads to an increase in KilR. KilR prevents proper FtsZ assembly by sequestering the protein subunits. The inhibition of cell division allows for greater repair leading to a decrease in mutation rates concomitant with an increase in the rate of recovery.

significant nucleotide conservation at specific regions (Appendix Fig S10A). These regions encompass the loops of the first and the middle hairpins and the downstream single-stranded RNA stretch (Appendix Fig S10B). In this study, we show that the middle hairpin B and the single-stranded region of OxyS interact with the *nusG* mRNA target of *E. coli*. As for *nusG*, conservation analysis of the interaction between OxyS and *nusG* predicts two possible base-pairing sites within *nusG* mRNAs: the first overlaps the *nusG* ribosome-binding site, and the second is located about 20–30 nt upstream of the Shine–Dalgarno sequence of *nusG*. Of 35 examples examined, 32 were predicted to harbor *nusG*/OxyS interaction; of which 14 were predicted to interact with both sites, and 17 were predicted to interact with the site overlapping the Shine–Dalgarno sequence and/or AUG of *nusG* or with the site located upstream (Appendix Fig S11). The OxyS sequence and structural conservation and the conservation of the responsive site in *nusG* indicate that regulation of *nusG* by OxyS is an important part in the bacterial response to oxidative stress. Coexistence of *nusG* with *kilR* or with λ *kil* gene is predicted in 16 of the respective examples, of which in 14 instances *nusG* is also predicted to base-pair with OxyS (Appendix Fig S12). Eighteen strains predicted to carry *nusG*/OxyS interaction lack *kilR* or λ *kil*. The effect of OxyS on *nusG*, a highly conserved and essential transcription factor in strains lacking Kil proteins, and the physiological impact this interaction may have remains to be addressed in future studies.

Small RNAs deploy a variety of mechanisms to cope with stress conditions such as, for example, reducing expression of outer membrane proteins in response to membrane stress, decreasing expression of iron-containing proteins in response to limiting iron or down-regulating the expression of sugar transporters, while up-regulating sugar de-phosphorylation in response to glucose–phosphate stress (Wagner & Romby, 2015). In all of these cases, the sRNAs cope with stress conditions by targeting genes directly linked to the specific stress. OxyS introduces a new regulatory concept in which a transient growth inhibition caused by this sRNA facilitates damage repair, thus enabling a higher number of cells to resume growth following stress. Whether additional sRNAs exploit the toxic coexistence of genes like *nusG* and *kilR* to coordinate cell growth with damage control remains to be addressed in further studies.

# Materials and Methods

### Bacterial growth conditions

*Escherichia coli* (MG1655) was grown at 37°C (200 rpm) in LB medium (pH 6.8). Ampicillin (100 μg/ml), tetracycline (10 μg/ml), chloramphenicol (20 μg/ml), and kanamycin (40 μg/ml) were added where appropriate. P*lac* and P*tac* promoters were induced with isopropyl β-D-thiogalactoside (IPTG; 1 mM) as indicated. (List of strains, plasmids, and DNA primers used in this study appears in Appendix Tables S1–S3.)

### Strain construction

Gene deletion mutants were generated using the gene disruption method as described (Yu *et al*, 2000). For construction of deletion mutants, tetracycline, chloramphenicol, or kanamycin, cassettes were amplified using mini-*Tn*10 chromosomal cassette and plasmids pKD3 or pKD4, respectively (Datsenko & Wanner, 2000). The PCR product (5–10 μg) purified using the Wizard SV PCR cleanup system (Promega, Madison, WI) was introduced into DY378 cells grown at 30°C to OD$_{600}$ of 0.4–0.6 and then transferred to 42°C for 15 min. The mutations were transferred into a fresh genetic background of MG1655 *mal*::*lacI*$^q$ by P1 transduction. The resistance cassettes were eliminated using pCP20 (Datsenko & Wanner, 2000). To construct Δ*lacZ*::*Tn*10, the chromosomal region flanked by genome coordinates 363264 and 366272 (GenBank entry NC_000913.3) was replaced by the *tetR* and *tetA* genes using primers 2081 and 2082. *lacZ* gene disruption was examined by PCR using flanking primers (2083 and 2084). To construct Δ*oxySli*::kan, the chromosomal region flanked by genome coordinates 4158322 and 4158375 (GenBank entry NC_000913.3) was replaced by the *kan* gene using primers 2169 and 2170. *oxyS* gene disruption was examined by PCR using flanking primers, 2026 and 2027. To construct Δ*kil*::*cat*, the chromosomal region flanked by genome coordinates 1417950 and 1418235 (GenBank entry NC_000913.3) was replaced by the *cat* gene using primers 2265 and 2295. *kil* gene disruption was examined by PCR using flanking primers 2267 and 2296. To construct NusG-SPA fusion in the chromosome, primers were designed to amplify the SPA tag together with the kanamycin resistance cassette from plasmid pJL148 (Zeghouf *et al*, 2004). At least 42 nt of homologous sequences was allowed. MDS42 *nusG*-SPA-*kan* was constructed using primers 2216 and 2217. The PCR products were gel-purified and then transform into MDS42 cells carrying pKD46 plasmid (Datsenko & Wanner, 2000). Insertions were confirmed by PCR using primers 2182 and 2219. The products were sequenced using primer 2227. The fusions were transferred into strains by P1 transduction.

### Plasmid construction

To construct P*lac-oxyS*, the *oxyS* sequence from its transcription start site and 27 nt downstream of its transcription terminator was PCR-amplified from MG1655 chromosomal DNA using primers 2026 and 2027 and cloned into the AatII and HindIII restriction sites of pBR-*plac* (Guillier & Gottesman, 2006). To construct P*tac-nusG-lacZ* translation fusion in the single-copy plasmid pBOG552 (Hershko-Shalev *et al*, 2016), the *nusG* fragment was PCR-amplified from MG1655 using primers 2179 and 2181 and then cloned into the EcoRI and SmaI sites of pKK177-3-*lacI*. The P*tac-nusG* fragment was sub-cloned into pBOG552 using the BamHI site. To construct P*lac-kilR*, *kilR* sequence was amplified from MG1655 chromosomal DNA using primers 2700 and 2701 and cloned into the AatII and HindIII restriction sites of pBR-*plac*. To construct P$_{secE}$ *secE-nusG*, the sequence of *secE-nusG* was PCR-amplified using primers 2377 and 2378 and then cloned into HindIII and BamHI restriction sites of pACYC184. To construct P$_{secE}$ *nusG*, we deleted the *secE* gene from the plasmid P$_{secE}$ *secE-nusG* using primers 2725 and 2727. To construct P$_{secE}$ *secE*, the sequence of *secE* was PCR-amplified using primers 2377 and 2727 and then cloned into HindIII and BamHI restriction sites of pACYC184. All pACYC (P15A) plasmids carry the OxyS complementary sequence. The cloning in pACYC plasmids destroyed the *tet* resistance cassette.

### Random mutagenesis

Random mutagenesis of OxyS wild type and OxyS$_{A69C}$ was carried out using hydroxylamine as described before (Hershko-Shalev *et al*, 2016).

### Site-directed mutagenesis

Mutations A69C, A69C,C70U, C56U,C58U, C76U,C77U, and C76G, C77G were generated by PCR using P*lac-oxyS* (pSA86) and two tail-to-tail divergent primers of which one carried the desired mutation. The PCR product was gel-purified and subjected to blunt end ligation. Likewise, the mutations in *nusG* were generated in P*tac-nusG* (pKK177-3). The mutated P*tac-nusG* fragment was sub-cloned into pBOG552 using the BamHI site.

### Survival assays

Overnight cultures of MG1655 strains as indicated in the legends, carrying control or OxyS plasmids, were grown from fresh transformation plates. Starters were diluted 1/100 in 20 ml LB (125-ml Erlenmeyer flasks) and grown at 37°C (200 rpm). IPTG (1 mM) was added at the time of dilution, where indicated H$_2$O$_2$ (1 mM) was added at OD$_{600}$ ~ 0.1 for 30 min. Thereupon, the cells were washed and suspended in fresh warm LB medium. Samples were taken at 30, 60, and 90 min after wash, diluted in 1× PBS, and plated. Each sample was plated twice. Colonies were counted, and the percentage of survival rate was calculated. Cultures with no *oxyS* plasmids were treated and assayed as above except for IPTG induction.

### β-galactosidase assays

Overnight cultures of MG1655 *relA*::*cat*, Δ*oxySli*::*frt*, *lacZ*::Tn10, *lacI*$^q$ carrying P*tac-nusG-lacZ* translational fusion (pSC101* *kan*$^r$) as well as P*lac* control and P*lac-oxyS* plasmids as indicated, were diluted 1/100 in 10 ml LB medium supplemented with ampicillin and kanamycin. The cultures grown to OD$_{600}$ ~ 0.1 were treated with IPTG (1 mM) to induce transcription of both *nusG-lacZ* and OxyS. β-Galactosidase activity was measured at 120 min after IPTG induction. To measure OxyS effects on λ*fhlA-lacZ* fusion, MG1655 λ*fhlA-lacZ*:*kan*, *relA*::*cat*, *lacI*$^q$, *lacZ*::Tn10 carrying P*tac-oxyS* wild type and mutants were diluted 1/100 in 5 ml LB medium supplemented with ampicillin. The cultures grown to OD$_{600}$ ~ 1.0 were treated with IPTG (1 mM) to induce OxyS expression. β-Galactosidase activity was measured at 60 min after IPTG induction.

### RNA extraction

To isolate total RNA, cultures grown as indicated in the specific legends were pelleted and resuspended in 50 μl of 10 mM Tris–HCl (pH 8) containing 1 mM EDTA. Lysozyme was added to 0.9 mg/ml, and the samples were subjected to three freeze–thaw cycles. Total RNA was purified using TRI reagent (Sigma) according to the manufacturer's protocol. For real-time analysis, the purified RNA was ethanol-precipitated in the presence of 0.3 M sodium acetate. To estimate the half-life of *secE-nusG* mRNA,

cultures of MG1655 *relA*::*cat, lacI*$^q$, and MG1655 *relA*::*cat, lacI*$^q$, Δ*oxySli*::*kan* grown to early-exponential phase in LB medium were treated with 1 mM hydrogen peroxide. After 2 min, rifampin (0.2 mg/ml) was added. Samples were taken at 2, 4, 6 min after the rifampin addition.

### Northern analysis

RNA samples (10–20 μg) were denatured for 10 min at 70°C in 98% formamide loading buffer, separated on 8 M urea–6% polyacrylamide gels, and transferred to Zeta-Probe GT membranes (Bio-Rad Laboratories) by electroblotting. To detect OxyS RNA, the membranes were hybridized with [$^{32}$P]-end-labeled *oxyS* primer (492) in modified CHURCH buffer (Church & Gilbert, 1984). To detect *secE-nusG* full-length RNA, samples (15 μg) were denatured for 10 min at 70°C in MOPS loading buffer, separated on 1.4% agarose gels, and transferred to Zeta-Probe GT membranes by capillary transfer. *secE-nusG* mRNA levels were detected using anti-*nusG*-labeled riboprobe synthesized using PCR template (2620 and 2554) as previously described (Hershko-Shalev *et al*, 2016). Riboprobe hybridization buffer contained 50% formamide, 3.5% SDS, 250 mM NaCl, 82 mM Na$_2$HPO$_4$, and 40 mM NaH$_2$PO$_4$ at pH 7.2. After 2 h at 50°C, the membranes were washed for 20 min at 50°C in 2× SSC, 1% SDS, then for 20 min at 55°C in 1× SSC, 0.5% SDS, and last for 20 min at 60°C in 0.5× SSC, 0.1% SDS. tmRNA (10Sa) was used as a loading control (primer 1912).

### Western of SPA-tagged *nusG*

Overnight cultures of SPA-tagged strains were diluted 1/100 and grown shaking (200 rpm) at 37°C in 100 ml LB (250-ml Erlenmeyer flasks). H$_2$O$_2$ (1 mM) was added at OD$_{600}$ ~ 0.1 for 2, 5, and 7 min. Samples were pelleted and then fluidized in 1× Laemmli sample buffer (Bio-Rad), heated at 95°C for 5 min, and centrifuged for 5 min. 10$^7$ cells of each sample were analyzed on SDS–PAGE (12%). The proteins were transferred to a nitrocellulose membrane (Sartorius); the blots were blocked with BSA and skim milk and probed with FLAG M2 monoclonal antibody (Sigma-Aldrich) according to the manufacturer's protocol. The membranes were also probed with Hfq-specific antibodies as a loading control. The proteins were visualized using secondary antibody Anti-Mouse IgG-Alkaline Phosphatase (FLAG) or Anti-Rabbit IgG-Alkaline Phosphatase (Hfq) (Sigma-Aldrich) according to the Alkaline Phosphatase development protocol (Promega).

### Real-time PCR

RNA concentrations were determined using a NanoDrop machine (NanoDrop Technologies). DNA was removed by DNase treatment according to the manufacturer's instructions (RQ1 RNase-free DNase, Promega). 2 μg of DNA-free total RNA was used for cDNA synthesis using MMLV reverse transcriptase and random primers (Promega). *kil* cDNA levels were analyzed by real-time PCR using specific primers (2531–2532) and SYBR green mix (Absolute SYBR GREEN ROX MIX, ABgene) with Rotor-gene 3000A (Corbett) according to the manufacturer's instructions. The level of 16S rRNA (*rrsB*; primers 1309–1310) (Park *et al*, 2007) was used to normalize *kil* levels. The relative amount of cDNA was calculated using the

standard curve method. A standard curve was obtained from PCR on serially diluted genomic DNA as templates and was analyzed using Rotor-gene analysis software 6.0.

### *In vitro* RNA synthesis

Purified PCR fragments of *nusG* (207 bp) and *oxyS* wild type and mutants (173 bp) generated using primers 2220–2221 and 2238–2027 were used as templates to produce *nusG* and OxyS RNAs of 176 nt and 109 nt, respectively. The RNAs were synthesized in 50 μl reactions containing T7 RNA polymerase (25 units; New England Biolabs), 40 mM Tris–HCl (pH 7.9), 6 mM $MgCl_2$, 10 mM dithiothreitol (DTT), 20 units RNase inhibitor (CHIMERx), 500 μM of each NTP, and 200 ng of purified PCR templates carrying the sequence of the T7 RNA polymerase promoter. To synthesized fully labeled RNA, 500 μM of each, UTP, GTP, CTP, and 40 μM ATP was used with 10 μCi [$^{32}$P]-ATP (specific activity 800 Ci/mmol). Synthesis was allowed to proceed for 2 h at 37°C and then terminated by heating (70°C, 10 min). To remove DNA template, 4 U of turbo DNase I (Ambion) was added (37°C, 30 min), followed by phenol/chloroform extraction and ethanol precipitation in the presence of 0.3 M ammonium acetate.

### Primer extension

Annealing mixtures containing in DEPC-treated water, 0.05 pmol of *in vitro*-synthesized *nusG* RNA, 0.6 pmol of 5′ end-labeled *nusG*-specific primer (2221) without or with 12 pmol of *in vitro*-synthesized OxyS RNAs (wild type and mutants) were heated for 10 min at 70°C and then chilled on ice for 20 min. Thereafter, the reactions were incubated at 37°C in 20 mM Tris–HCl, 10 mM magnesium acetate, 0.1 M $NH_4Cl$, 0.5 mM EDTA, 2.5 mM β-mercaptoethanol, and 0.5 mM each dNTP for 15 min, at which reverse transcriptase (Promega; 40 units) was added. cDNA synthesis was allowed to proceed for 10 min at 37°C. The extension products were separated on 6% sequencing gels, alongside with sequencing reactions.

### RNase protection assay

0.06 pmol of labeled *nusG* RNA and 15 and 30 of OxyS RNAs were precipitated and annealed as described before (Rio *et al*, 2011). After overnight annealing at 45°C, the mixture was subjected to ribonuclease (A and T1) degradation at room temperature for 10 min. The products were separated on 8% sequencing gels, alongside with labeled RNA Marker (Decade Thermo Fisher Scientific).

### EMSA

Fully labeled *in vitro*-synthesized wt and quadruple mutant *nusG* RNAs (176 nt, 0.01 pmol, and 0.004 pmol, respectively) without and with increasing concentrations of OxyS (5, 10, and 15 pmol left panel, or 2, 4 right panel) were incubated for 15 min at 85°C. Thereafter, the mixtures were incubated at 37°C or 42°C for 60 min in binding buffer (6.7 mM Tris-acetate (pH 7.4), 3.3 mM Na-acetate, 10 mM DTT, and 10 mM $MgCl_2$). The RNA samples were separated on 4.5% nondenaturing polyacrylamide gels (19:1) in 20 mM Tris–HCl (pH 7.5), 60 mM KCl, and 10 mM $MgCl_2$ (50 volts for 5–6 h at 4°C).

### Mutagenesis assays

Overnight cultures carrying OxyS plasmids wild type and mutant with or without a plasmid expressing *ftsQAZ* (pFtsZ) were diluted 1/100 in 7 ml LB medium supplemented with ampicillin (for *oxyS* plasmids) and tetracycline (for pFtsZ). The cultures were grown to $OD_{600} \sim 0.1$ (37°C, 200 rpm) prior to the addition of IPTG (1 mM). At 15 min of IPTG induction, the cultures were exposed to 5 mM hydrogen peroxide at room temperature (22°C), without shaking for 15 min. Thereupon, 0.5 ml of cells was mixed with 2 ml fresh LB and grown overnight (23 h). To determine frequencies of mutagenesis, aliquots were taken after 23 h and plated on LB plates containing 100 μg/ml of rifampicin. The numbers of Rif$^r$ mutants were normalized to the numbers of viable cells at the 23-h time point. To estimate the number of mutations due to KilR expressed from P*lac*-*kilR* plasmid, cells were treated as above, except that expression of *kilR* was induced using 20 μM of IPTG instead of 1 mM. To estimate the number of mutations in wild type and *oxyS* mutant, cultures at $OD_{600} \sim 0.1$ were exposed to 1 mM hydrogen peroxide for 30 min (37°C, 200 rpm). The numbers of Rif$^r$ were calculated following the protocol described above.

### Fluorescence microscopy

Overnight cultures were sub-cultured in fresh LB medium supplemented with 1 mM IPTG and Amp. Tetracycline was included only in cultures that expressed pFtsQAZ. Cultures were grown at 37°C for 3 h before imaging. All the samples were spotted on PBS agar pad for imaging. DNA was stained with DAPI (Sigma-Aldrich) at a final concentration of 2 μg/ml. Cells were visualized and photographed using Nikon Eclipse Ti-E inverted microscope equipped with Perfect Focus System (PFS) and ORCA Flash 4 camera (Hamamatsu photonics). Images were processed using NIS Elements-AR software. To determine cell length distribution, > 250 cells from a 4 × 4 matrix field (16 individual fields) were randomly selected, and their length was measured using NIS Elements-AR software. The values given are data obtained from three independent experiments, a total of > 750 cells.

### Computational prediction of OxyS homologs and targets

Iterative blasting and structural filtering implemented in the GLASSgo web server (http://rna.informatik.uni-freiburg.de/GLASSgo/Input.jsp) was used to detect potential OxyS homologs. The maximum allowed *E*-value was set to 1, the minimum required identity was 58%, and the structure-based filtering value was set to 2. This search yielded 603 candidates belonging to 10 different genera and some uncharacterized enterobacterial sequences, representing at least 27 different species and more than 500 different strains. The computational OxyS target prediction was conducted using IntaRNA and CopraRNA (Wright *et al*, 2014) on the web server version with standard parameters.

**Expanded View** for this article is available online.

### Acknowledgements

This work was supported by the German-Israeli Foundation (G-1311-416.13/2015); the Israel Science Foundation founded by The Israel Academy of

Sciences and Humanities (711/13); the Israel Centers of Research Excellence (ICORE), Chromatin and RNA (1796/12); the German Federal Ministry for Education and Research (BMBF) program de.NBI-Partner (Grant 031L0106B) and Deutsch-Israelische Projektkooperation (AM 441/1-1 SO 568/1-1).

## Author contributions

SB, ME-W, JE, JG, SG, MH, and PRW performed the experiments. SB, ME-W, JE, JG, SG, MH, PRW, SA, and WRH conceived the experiments and analyzed the data. SA wrote the manuscript. SA and WRH managed the project.

## Conflict of interest

The authors declare that they have no conflict of interest.

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
