## [Review Process File · The EMBO Journal]

OxyS small RNA induces cell cycle arrest to allow DNA damage repair

Shir Barshishat, Maya Elgrably-Weiss, Jonathan Edelstein, Jens Georg, Sutharsan Govindarajan, Meytal Haviv, Patrick R. Wright, Wolfgang R. Hess and Shoshy Altuvia

Review timeline:

Submission date:	22 June 2017
Editorial Decision:	14 July 2017
Revision received:	24 October 2017
Editorial Decision:	10 November 2017
Revision received:	13 November 2017
Accepted:	20 November 2017

Editor: Hartmut Vodermaier

Transaction Report:

1st Editorial Decision

14 July 2017

Thank you for submitting your manuscript on OxyS-mediated cell cycle arrest for our consideration. We have now received comments from three expert reviewers, copied below for your information. As you will see, all referees consider your findings interesting and potentially important, but they also raise a number of substantive issues that would require improvement before eventual publication. Some of these concerns (especially those listed by referees 1 and 2) pertain primarily to the genetic analyses and the OxyS-nusG RNA connection, and it would clearly be essential to satisfactorily address these issues (in particular, major point 1 of referee 2) with further experiments. The other major criticisms relates to the fact that the proposed model remains very inferential downstream of the demonstrated OxyS-NusG-Kil axis, especially with regard to the functional significance of OxyS-dependent Kil expression and cell division inhibition for protection from DNA damage and mutagenesis (as detailed by referee 3). In our view, addressing these mechanistic aspects through decisive additional data along the lines suggested by the referee would indeed be essential to warrant publication as an EMBO Journal article, but I do realize that this may require substantial further time and efforts, and also be of unclear outcome. I have therefore taken the liberty to briefly discuss the study and the reviews with my colleague Dr. Esther Schnapp from our sister journal, EMBO Reports. Esther also found the study interesting, and in principle suitable for EMBO Reports even without further mechanistic extensions, as long as the respective points of referee 3 would be at least diligently discussed, and the more fundamental concerns of referees 1 and 2 fully addressed.

Given the overall interest of the work, I would thus like to invite you to revise the manuscript in response to the referee reports. Should you be prepared to deepen and expand the mechanistic experiments as requested by referee 3 (if necessary within an extended revision duration), we would be happy to pursue publication of the revised study in The EMBO Journal; a link below will allow you to resubmit the revision in this case. Should you however prefer to rapidly publish this work pending only the substantiation requested by referees 1 and 2, then I would encourage you to transfer the manuscript to EMBO Reports directly using the second link given below, and to get in contact with my colleague (esther.schnapp@embo.org) to discuss further proceedings.

I should add that in either case, our policy to allow only a single round of major revision will make it important to carefully and comprehensively respond to all points raised during this round. Furthermore, as a matter of EMBO Press policy, competing manuscripts published elsewhere during the revision for either journal would have no negative impact on our final assessment of your revised study. Additional information on preparing and uploading a revision can be found below.

Thank you again for the opportunity to consider this work for publication, and please feel free to contact me with any questions about submission of the revised manuscript to either The EMBO Journal or EMBO Reports. We look forward to hearing from you.

REFEREE REPORTS

Referee #1:

In this manuscript Barshishat et al. address a long-standing question in the field of bacterial RNA biology, i.e. what is the molecular mechanism underlying DNA damage protection by the OxyS small regulatory RNA (sRNA)? OxyS was one of the first bacterial sRNAs discovered and has served as a model for mechanistic analysis on sRNA functions thereafter (see Zhang et al 1998, EMBO J. and Zhang et al 2002, Molecular Cell). This being said, studies focusing on the biological role of OxyS have been lagging behind and key question such as the role of OxyS for cell survival during exposure to DNA damaging agents have not been addressed. This manuscript addresses this question starting from a set of OxyS mutants displaying increased or decreased toxicity when ectopically over-expressed. Bioinformatic analysis of potential OxyS target genes that could explain this phenotype predicted base-pairing of OxyS with the 5' end of the nusG mRNA, encoding a modulator of transcription termination in *E. coli* and related bacteria. The authors show that OxyS-mediated repression of nusG leads to the production of Kil protein, which in turn associates with components of the cell division machinery to inhibit cell separation. Indeed, OxyS overexpression results in elongated *E. coli* cells, a phenotype that is suppressed when cells lack the kil gene. Moreover, lack of oxyS or kil affects cell recovery rates after exposure to DNA damaging agents. Together, this manuscript describes an interesting new mechanism of how regulatory RNA can interfere with cell division and survival by modulating a core regulatory process such as transcription termination. Overall, the rationale and interpretation of the experiments is sound, however, I have some comments that should be addressed prior to publication of this manuscript.

Major points of criticism:

- Page 6, 2nd paragraph and Fig. 2: nusG is the second gene of an operon with secE. Does OxyS also regulate secE and is that relevant for the phenotype(s) observed?
- Page 6, 2nd paragraph: the authors mention that pyrG and orn could be additional targets of OxyS. Have these been tested for regulation and, if so, does regulation of either of the targets affect toxicity?
- Page 7, 3rd paragraph: I don't fully understand the use of the oxyS^{li::kan} mutant. Is it relevant for these experiments to keep the 5' and 3' end of the chromosomal oxyS gene intact?
- Figure 3F: the contrast on the western blot images is very poor and does not seem to reflect the quantification of the data above.
- Figure 4A: It would be useful to include one panel showing the effect of kil expression on cell shape.
- I think Fig. 6 is not essential for the manuscript and could be moved to the SI.
- The main manuscript does not seem to contain a "Materials and Methods" section.

Minor points of criticism:

- Fig. 2D: I think the figure legend and the main text should state more clearly that these experiments were performed in vitro. Also, why is there no decrease in the levels of the full-length cDNA when OxyS is added to the reaction?
- Fig. 3D, E: I think it would be useful to plot these data on a semi-log graph to follow the decay of nusG transcript. Also, is this the nusG transcript only, or secE-nusG?
- Figure 4C: why are the kil deficient mutants elongated in the absence of stress?
- Page 12, 2nd paragraph, 3rd sentence: The connection of OxyS mediated toxicity and RelA seems more of a hypothesis at this point. Therefore, this sentence should be worded more carefully.
- Page 13, last sentence: "biomolecules" is not very specific - please rephrase.
- Throughout the figures: italicize gene names.

Referee #2:

This paper deals with the regulatory function of a well-known small regulatory RNA in *Escherichia coli*, OxyS. Although the regulon of this sRNA has been studied in details, the mechanism by which OxyS protected cells from DNA damage upon oxidative stress was still not yet defined. Using a combination of approaches, the authors demonstrated that OxyS temporally affected cell growth in an indirect manner to prevent DNA damages and/or to allow DNA damage repair following stress. The key experiment was the selection of suppressor mutations in OxyS that restore cell growth upon induction of oxidative stress. Surprisingly the authors have discovered a cascade of events, which affected the polymerization of FtsZ to induce cell growth arrest. The first event involves the repression of nusG mRNA translation mediated through the formation of conserved base pairings with OxyS sRNA. In turn, the repression of the transcription termination factor NusG leads to the synthesis of the Kil protein, produced from a cryptic prophage, that is known to interfere with the polymerization of FtsZ. This work shows an unexpected feature of sRNA-dependent regulation that involves both stress response and protection against DNA damage. One may expect that this finding is more general rather than an exception. Moreover this study brings another demonstration of the positive impact of the mobile elements on bacterial physiology, and of the crosstalks with the core genome to adapt bacterial growth upon stress.

Major comment:

1- To probe the interaction between wild type and mutated OxyS to nusG mRNA, the authors use primer extension. Although the data support the author claims, this is not the best method to map RNA-RNA pairings. Indeed, many studies have shown that not all RNA-RNA pairings are able to stop RT extension. Pauses are more often observed at GC rich pairings. Therefore, the absence of RT pause with OxyS C76GC77G does not necessarily mean that binding of the mutated OxyS to nusG mRNA does not occur. Indeed, the mutant C76UC77U OxyS is still able to induce RT pause although the repression of nusG-lacZ fusion is lost. The authors should use a more direct approach such as gel retardation assays, which would be more precise to monitor the effect of the mutations in OxyS on the stability of the duplexes formed with nusG mRNA. The β -galactosidase assays were not performed with OxyS C76GC77G (Fig. 2C), and mutations in nusG mRNA would have been appropriate to restore the base pairings with OxyS C76GC77G, in order to fully demonstrate that the regulation results from sRNA-mRNA pairings.

Minor comments:

1- Figure 3D, E: the deletion of oxyS has only a minor effect on the half life of nusG mRNA. It would be better to show the graph of the % of mRNA remaining versus time in the wild type and mutant strains. Figure 3F: the effect of OxyS expression in the wild type strain has also a minor effect on NusG-SPA synthesis. In addition, why the expression of NusG-SPA is significantly enhanced upon stress conditions in Δ oxyS mutant strain? Northern analysis would have been appropriate to monitor OxyS expression in the wild type strain. I assume that the expression of

OxyS in the WT strain is much more lowered than the expression of OxyS from the plasmid as shown in Figure 2B, C. Because most of the phenotypes have been observed under the overexpression of OxyS (Fig. 4A, B), this should be clearly mentioned and discussed.

2- In support to the proposed model of regulation, the interaction between NusG-OxyS has been found conserved in many bacteria as well as the co-existence of NusG-OxyS and Kil. However, there are several examples showing that base pairings between OxyS and nusG mRNA were not predicted whether Kil was predicted or not. For instance in several strains such as *Citrobacter freundii*, Enterobacteriaceae FGI-57 or *Shigella dysenteriae*, both NusG and Kil were predicted but pairings between OxyS and nusG mRNA were not detected. The message should be modified in the discussion taking into account these various situations.

Referee #3:

This paper helps to solve a long-standing question about the small stress-induced OxyS RNA of *E. coli*: how does OxyS protect cells against DNA damage? In seeming contradiction to the protective effects of OxyS, the authors found that overexpression of OxyS is toxic and induces cell filamentation, indicating that OxyS inhibits cytokinesis. As SOS systems that respond to DNA damage also inhibit cytokinesis, the authors hypothesized that perhaps OxyS induces a SOS-like cell division delay that protects cells against stress by allowing time for repair. Their data are consistent with a model in which OxyS is induced upon oxidative stress (Altuvia et al. 1997), and inhibits expression of nusG by antisense base pairing with its mRNA, which leads to derepression of expression of kil on the prophage rac by antitermination. They propose that Kil then binds to the cell division protein FtsZ and inhibits its assembly, analogous to the effect of phage lambda Kil, causing the observed cell division delay.

The strength of the paper is the solid genetic evidence for the basic aspects of the above new mechanism, and other than some glitches with English, the core of the work is well presented. However, given that this is a manuscript for EMBO J., I was disappointed in the relatively superficial way the mechanism was addressed and discussed.

First, there is no evidence or even discussion about how such a cell division delay might be transient. For example, might Kil be unstable, by analogy with Sula, the SOS-induced inhibitor of FtsZ?

Second, excess OxyS reduces growth rate (based on cell density increase), yet these cells are all highly filamentous under the microscope, indicating that they are still growing robustly while no longer dividing. How much of the effect on cell density increase is due to off-target effects that have nothing to do with cell division inhibition?

Third, there is no real attempt to explain the anti-mutagenic properties of OxyS, other than to propose that the cell division delay somehow inhibits DNA damage/mutagenesis. Near the bottom of page 11 in the Discussion, the authors state that "OxyS leads to a significant reduction in DNA damage and an increase in the rate of recovery". Is there direct evidence that there is a significant reduction in DNA damage, or is this inference only based on the rif-resistance data in Table 1? Isn't a major effect of Sula-mediated delay in cell division to prevent double-strand breaks caused by guillotining of the nucleoid by a closing cell division septum?

Fourth, the authors incorrectly conflate the known mechanism behind the Kil protein of phage Lambda, which has been shown to directly interact with FtsZ and affect its polymerization properties, with the unknown mechanism behind the Kil protein of prophage Rac. The latter is known to affect FtsZ rings (Conter, Bouche and Dassain, 1996), but anything more mechanistic is pure conjecture given that Rac Kil and Lambda Kil share little sequence identity.

Finally, it would have been a satisfying confirmation of their model if transient expression of Rac Kil from a plasmid, which would bypass NusG signaling, had some benefits for stress recovery. This would be particularly useful in the cell biology experiments, as the persistence of ZapA-GFP ring/spiral structures in the filaments suggests that Rac Kil may act via a different mechanism than Lambda Kil, perhaps at a later step in cell division that extra FtsA+FtsZ could still correct.

Other comments:

- 1) Page 6: Is there any reason why *pyrG* and *orn* might not also be bona fide targets of OxyS?
- 2) Page 8 first paragraph and Fig. 3A: the recovery rate of the nontoxic OxyS mutant is between the rate of the vector control and OxyS, not similar to that of the control cells as stated in the text.
- 3) Page 9: The authors should rationalize why they used ZapA-GFP instead of FtsZ-GFP. Presumably it is because they were concerned that expression of *ftsZ-gfp* might suppress the effects of Kil. However, *zapA-gfp* expression may have the same effect to some extent, as it can bundle FtsZ polymers and protect against FtsZ assembly inhibitors. Did the authors try immunofluorescent staining using anti-FtsZ to avoid perturbing levels of cell division proteins?
- 4) In Fig. 4A, it looks like ZapA-GFP forms fairly coherent structures at somewhat regular intervals. Assuming that these reflect FtsZ, it is not necessarily the case that "ring assembly is impaired", stated on page 9, although it is clear that the structures cannot progress further in cytokinesis.
- 5) In Fig. 4B, a significant fraction of cells are still filamentous after *oxyS* expression even in the absence of kil. Does that implicate alternative Kil-like proteins such as DicB, which may also be induced by OxyS?
- 6) Fig. 4C: why does kil expression decrease more than 2-fold in the first 15 minutes after peroxide addition?
- 7) MC4100 already has a *relA1* allele, so why was it necessary to make a *relA::kan* or *relA::cat* derivative?
- 8) MC4100 also has a *spoT1* mutation (Spira, Hu & Ferenci, Microbiology 2008) that should render it devoid of ppGpp. Is such a strain a valid model for a typical response to stresses?
- 9) Whenever the *pftsQAZ* plasmid is used, the authors assume that the effects they see are due to FtsZ. While this is likely true, they cannot rule out effects mediated by extra FtsA (or FtsQ?) as well, unless they express FtsZ by itself.
- 10) Page 10, top, and Fig. 4C: as mentioned above, the population, albeit reduced, of significantly longer cells even in the absence of OxyS or Kil suggests that there are alternative pathways to help cells exposed to oxidative stress survive that involve cell division arrest.

Minor English/typo issues:

- 1) Page 3 line 6: this sentence would be clearer after deleting the phrase "the expression of which is".
- 2) Page 5 line 14: change to "harmless clustered in two sites:"
- 3) Page 6 line 5: change to "targets whose putative complementary sites might match the changes."
- 4) Page 7 line 14: delete "a"
- 5) Page 8 line 3: delete comma
- 6) Page 8: there is inconsistent use of present and past tense, particularly in this paragraph.
- 7) Page 9, line 17: change to "fusions"
- 8) Page 13, lines 7-8: change to "we show that the middle hairpin B and the single-stranded region of OxyS interact..."
- 9) Fig 3C, y-axis should define it as "kil mRNA copy number"
- 10) Fig. 4C: y-axis: "Length" is misspelled

October 24, 2017

Dear Dr. Vodermaier,
Senior editor / The EMBO Journal

Re: Revised manuscript EMBOJ-2017-97651

We wish to thank all the referees for their useful suggestions. To address the referee's comments, we carried out additional experiments. The main manuscript now includes new data. Below we present a point-by-point response to their comments.

Referee #1

Major points of criticism:

1. Page 6, 2nd paragraph and Fig. 2: *nusG* is the second gene of an operon with *secE*. Does OxyS also regulate *secE* and is that relevant for the phenotype(s) observed?

Reply: The OxyS-dependent decrease of the full-length *secE-nusG* mRNA was detected on northern blots of agarose gels using a *nusG* specific riboprobe. Thus, it is possible that in the presence of OxyS, *secE* is decreased. However, the decrease in *secE* is not relevant to the toxic phenotype of OxyS. OxyS is less toxic in the presence of a plasmid expressing *nusG*, whereas in the presence of a plasmid expressing *secE*, *oxyS* is highly toxic (new Appendix Fig. S3). These results indicate that an increase in the expression levels of *nusG* can negate *oxyS* toxicity, whereas a concomitant increase in *secE* has no effect on *oxyS* toxicity. Therefore, *oxyS* is toxic because it decreases *nusG* expression levels. Page 8

2. Page 6, 2nd paragraph: the authors mention that *pyrG* and *orn* could be additional targets of OxyS. Have these been tested for regulation and, if so, does regulation of either of the targets affect toxicity?

Reply: Subsequent to submission, we examined the effect of *oxyS* on *pyrG* and *orn* using riboprobes specific to these genes. Unfortunately, under the conditions tested we could not detect *orn* or *pyrG* mRNAs at all.

3. Page 7, 3rd paragraph: I don't fully understand the use of the *oxySli::kan* mutant. Is it relevant for these experiments to keep the 5' and 3' end of the chromosomal *oxyS* gene intact?

Reply: Since the promoter of *oxyS* overlaps the *oxyR* promoter and to avoid any possible polar effects we left the promoter and the transcription termination signal of *oxyS* intact. *oxySli* is a clean mutant in which only the regulatory elements relevant for OxyS toxicity were deleted. This is now mentioned in the text. Page 9

4. Figure 3F: the contrast on the western blot images is very poor and does not seem to reflect the quantification of the data above.

Reply: We replaced the images with higher quality images; we hope this improves the presentation. As for the quantification, relative intensity denotes the levels of NusG-SPA relative to Hfq. This is now clearly indicated in the figure. Former Figure 3F is now Figure 4F.

5. Figure 4A: It would be useful to include one panel showing the effect of *kil* expression on cell shape.

Reply: *kilR* gene of the cryptic phage *rac* was cloned under and expressed from the *Plac* promoter. Expression of *kilR* results in cell filamentation. This image is now presented in Figure 5A. Former Figure 4A is now Figure 5A.

6. I think Fig. 6 is not essential for the manuscript and could be moved to the SI.

Reply: As suggested Fig. 6 is now Appendix Fig. S12

7. The main manuscript does not seem to contain a "Materials and Methods" section.

Reply: The revised manuscript includes this section

Minor points of criticism:

1. Fig. 2D: I think the figure legend and the main text should state more clearly that these experiments were performed *in vitro*. Also, why is there no decrease in the levels of the full-length cDNA when OxyS is added to the reaction?

Reply: The text and the legend were corrected as suggested. As for the decrease in the levels of the full-length cDNA, we suspect that it is a technical loading problem. To obtain accurate values of termination vs. full length, we calculated the ratio of termination signals per full-lengths. Wild type *oxyS-nusG* interaction was used as a 100% reference. The relative intensity is now indicated in Fig. 2D

2. Fig. 3D, E: I think it would be useful to plot these data on a semi-log graph to follow the decay of *nusG* transcript. Also, is this the *nusG* transcript only, or *secE-nusG*?

Reply: We changed the presentation of these data as suggested. Former Figure 3D is now Figure 4E.

The OxyS-dependent decrease of the full-length *secE-nusG* mRNA was detected on northern blots of agarose gels using a *nusG* specific riboprobe. Thus, it is possible that *secE* is decreased. In any case, the decrease in *secE* is not relevant for the toxic phenotype of OxyS. OxyS is less toxic in the presence of a plasmid expressing *nusG*, whereas in the presence of a plasmid expressing *secE*, *oxyS* is highly toxic (new Appendix Fig. S3). These results indicate that an increase in the expression levels of

nusG can negate *oxyS* toxicity, whereas a concomitant increase in *secE* has no effect on *oxyS* toxicity. Therefore, *oxyS* is toxic because it decreases *nusG* expression levels. Page 8.

3. Figure 4C: why are the *kil* deficient mutants elongated in the absence of stress?

Reply: **Per** this question, we revisited our presentation of the data in Fig. 4C, in which we used mean values. Mean value has the disadvantage of being affected by any extreme value compared to the rest of the samples. Therefore, we think that in this case, median values are more appropriate, and a better measure of the mid point. Accordingly, we replaced Fig 4C mean for median (current Figure 5C). It is important to note and the statistics is identical.

4. Page 12, 2nd paragraph, 3rd sentence: The connection of OxyS mediated toxicity and RelA seems more of a hypothesis at this point. Therefore, this sentence should be worded more carefully.

Reply: The sentence has been corrected as suggested

5. Page 13, last sentence: "biomolecules" is not very specific - please rephrase.

Reply: The sentence has been corrected as suggested.

6. Throughout the figures: italicize gene names.

Reply: Corrected as indicated

Referee #2

Major comment:

1. To probe the interaction between wild type and mutated OxyS to *nusG* mRNA, the authors use primer extension. Although the data support the author claims, this is not the best method to map RNA-RNA pairings. Indeed, many studies have shown that not all RNA-RNA pairings are able to stop RT extension. Pauses are more often observed at GC rich pairings. Therefore, the absence of RT pause with OxyS C76GC77G does not necessarily mean that binding of the mutated OxyS to *nusG* mRNA does not occur. Indeed, the mutant C76UC77U OxyS is still able to induce RT pause although the repression of *nusG-lacZ* fusion is lost. The authors should use a more direct approach such as gel retardation assays, which would be more precise to monitor the effect of the mutations in OxyS on the stability of the duplexes formed with *nusG* mRNA. The β -galactosidase assays were not performed with OxyS C76GC77G (Fig. 2C), and mutations in *nusG* mRNA would have been appropriate to restore the base pairings with OxyS C76GC77G, in order to fully demonstrate that the regulation results from sRNA-mRNA pairings.

Reply: We thank the reviewer for this comment. Indeed, we found that *oxyS/nusG* interaction is more complex. The revised manuscript now includes experimental data indicating that two molecules of OxyS can bind one *nusG* molecule at two different sites. The first binding site spans nucleotides -40 to -29 with respect to the first nucleotide of the start codon of *nusG*. The second experimentally verified interaction site in *nusG* spans from nucleotide -20 to +13 with respect to the AUG, overlapping the RBS of *nusG*. We show that the expression of NusG quadruple mutant (G-15C; G-16C and G-31C; G-32C) carrying mutations at the two sites predicted to bind the same sequence in OxyS is repressed by OxySC76G; C77G and unaffected by wild type OxyS. Together the data indicate that two molecules of OxyS can bind two different sites in *nusG*.

Also, we now include an RNase protection and a gel retardation assay as more direct approaches to monitor binding OxyS/*nusG* binding (Appendix Fig. S2A and B).

Minor comments:

1. Figure 3D, E: the deletion of *oxyS* has only a minor effect on the half life of *nusG* mRNA. It would be better to show the graph of the % of mRNA remaining versus time in the wild type and mutant strains.

Reply: As suggested, we now present the data as % of mRNA remaining (current Figure 4E).

2. Figure 3F: the effect of OxyS expression in the wild type strain has also a minor effect on NusG-SPA synthesis. In addition, why the expression of NusG-SPA is significantly enhanced upon stress conditions in $\Delta oxyS$ mutant strain? Northern analysis would have been appropriate to monitor OxyS expression in the wild type strain. I assume that the expression of OxyS in the WT strain is much more lowered than the expression of OxyS from the plasmid as shown in Figure 2B, C. Because most of the phenotypes have been observed under the overexpression of OxyS (Fig. 4A, B), this should be clearly mentioned and discussed.

Reply: We monitored OxyS expression in wild cells exposed to hydrogen peroxide and compared its levels to OxyS expression from *Plac-oxyS* plasmid upon induction with IPTG. At 10 and 30 min of H₂O₂ induction, the levels of OxyS were about 9 and 6.5-fold lower than OxyS levels produced by the plasmid (new Appendix Fig. S4). Page 9

The effect of the chromosomally encoded *oxyS* allele on NusG levels is indeed minor. Yet, it is important to note that most sRNAs were shown to affect their targets when overexpressed. Here we show that the chromosomally encoded *oxyS* allele modulates the expression of its target in response to oxidative stress. More importantly, OxyS renders the cells more resilient to the stress; the survival rate of cells expressing the chromosomally encoded *oxyS* allele is significantly higher than the rates detected with *oxyS* mutant. In addition, NusG is a fundamental transcription factor that modulates RNAP processivity and termination properties. NusG can also bind ribosomal protein

S10 (NusE). This interaction, mutually exclusive with the Rho interaction, provides a physical framework for the coupling of transcription and translation. Given the functional characteristics of this protein, we suspect that even a small effect on *nusG* expression is critical, affecting bacterial metabolism.

As for the stress-dependent increase in NusG-SPA levels in $\Delta oxyS$ mutant – in the absence of data related to *nusG* expression control, we hesitate to speculate. In *E. coli*, NusG protein was found to be associated with thioredoxin (Kumar et al., PNAS 2004). Thioredoxin is known to participate in protection against H₂O₂ by scavenging reactive oxygen species and by regulating the activity of detoxification proteins. The association of NusG with thioredoxin is unclear. In *Caulobacter crescentus*, the stationary phase induced response regulator SpdR was found to modulate expression of a number of genes including NusG and Hfq (da Silva et al., BMC Microbiology 2016). Numerous studies on sRNA-mRNA interactions carried out in *E. coli* and *Salmonella* have demonstrated that master regulators are often subjected to regulation by multiple sRNAs. In light of these observations, it is possible that the stress-dependent increase in NusG-SPA levels in $\Delta oxyS$ mutant results from an effect of other sRNAs and/or protein factors.

2. In support to the proposed model of regulation, the interaction between NusG-OxyS has been found conserved in many bacteria as well as the co-existence of NusG-OxyS and Kil. However, there are several examples showing that base pairings between OxyS and *nusG* mRNA were not predicted whether Kil was predicted or not. For instance in several strains such as *Citrobacter freundii*, Enterobacteriaceae FGI-57 or *Shigella dysenteriae*, both NusG and Kil were predicted but pairings between OxyS and *nusG* mRNA were not detected. The message should be modified in the discussion taking into account these various situations.

Reply: Our experiments indicate that two independent molecules of OxyS bind to one *nusG* molecule. The first binding site spans nucleotides -40 to -29 with respect to the first nucleotide of the start codon. The respective binding site in OxyS is strongly conserved in all investigated homologs. Using IntaRNA2, this interaction is the optimal interaction in many investigated organisms. The second experimentally verified interaction site (predicted by IntaRNA1) in *nusG* spans nucleotides -20 to +13 with respect to the start codon. A new search taking into consideration these two experimentally verified sites showed that of 35 examples examined, 32 were predicted to harbor *nusG*/OxyS interaction; of which coexistence of *nusG* with *kilR* or with λkil gene was predicted in 14 of the respective examples (Appendix Fig. S12). These various situations are now discussed in the text. Page 16

Referee #3

1. There is no evidence or even discussion about how such a cell division delay might be transient. For example, might Kil be unstable, by analogy with SulA, the SOS-induced inhibitor of FtsZ?

Reply: In the revised manuscript, we show that by 60 minutes of exposure to hydrogen peroxide, the steady state levels of OxyS become extremely low. In Figure 4C we show that the levels of *kilR* reach a plateau at about the same time. Together these results indicate that the *oxyS/nusG/kilR* dependent cascade of events is transient. This is now discussed in the text. Page 14

2. Excess OxyS reduces growth rate (based on cell density increase), yet these cells are all highly filamentous under the microscope, indicating that they are still growing robustly while no longer dividing. How much of the effect on cell density increase is due to off-target effects that have nothing to do with cell division inhibition?

Reply: The phase contrast images presented in the current Figure 5 shows that the majority, if not all cells, are filamentous. We suspect that not all filaments are viable; the samples taken for imaging are routinely washed twice prior to visualization. Conceivably, dead/lysed cells were excluded from the sample. Furthermore it is unclear how many viable cells are produced per one filamentous cell, which could explain the difference between growth and reduced survival rate.

3. There is no real attempt to explain the anti-mutagenic properties of OxyS, other than to propose that the cell division delay somehow inhibits DNA damage/mutagenesis. Near the bottom of page 11 in the Discussion, the authors state that "OxyS leads to a significant reduction in DNA damage and an increase in the rate of recovery". Is there direct evidence that there is a significant reduction in DNA damage, or is this inference only based on the rif-resistance data in Table 1? Isn't a major effect of SulA-mediated delay in cell division to prevent double-strand breaks caused by guillotining of the nucleoid by a closing cell division septum?

Reply: Indeed this phrasing has been inaccurate. We propose that by inhibiting cell division OxyS enables the bacterial repair systems to repair the damage for a longer period of time. We rephrased the text accordingly. Pages 13 and 14

4. The authors incorrectly conflate the known mechanism behind the Kil protein of phage Lambda, which has been shown to directly interact with FtsZ and affect its polymerization properties, with the unknown mechanism behind the Kil protein of prophage Rac. The latter is known to affect FtsZ rings (Conter, Bouche and Dassain, 1996), but anything more mechanistic is pure conjecture given that Rac Kil and Lambda Kil share little sequence identity.

Reply: We thank the reviewer for highlighting this issue. We have corrected the text accordingly. In the revised manuscript, we distinguish between lambda *kil* and *rac kilR*.
Pages 13

5. It would have been a satisfying confirmation of their model if transient expression of Rac Kil from a plasmid, which would bypass NusG signaling, had some benefits for stress recovery. This would be particularly useful in the cell biology experiments, as the persistence of ZapA-GFP ring/spiral structures in the filaments suggests that Rac Kil may act via a different mechanism than Lambda Kil, perhaps at a later step in cell division that extra FtsA+FtsZ could still correct.

Reply: We are grateful to the reviewer for highlighting this important issue. Inadvertently, we carried out the experiments presented in Fig. 4A at 30°C, a suboptimal growth temperature. We repeated these experiments at 37°C and found that under optimal growth conditions, in cells expressing toxic OxyS, *zapA-gfp* is diffused, forming no structures, indicating that ring assembly is indeed impaired. The revised manuscript includes a new panel A in Fig 5 (former Figure 4).

We also show that cells with in *trans* expression of Rac KilR from a plasmid exhibit a low number of Rif^r mutants similar to the results obtained with OxyS expression. This indicates that NusG signaling can be bypassed by in *trans* expression of KilR. Page 11 and Table 1

Other comments:

1) Page 6: Is there any reason why *pyrG* and *orn* might not also be bona fide targets

Reply: Subsequent to submission, we examined the effect of *oxyS* on *pyrG* and *orn* using riboprobes specific to these genes. Unfortunately, under the conditions tested we could not detect *orn* or *pyrG* mRNAs at all.

2) Page 8 first paragraph and Fig. 3A: the recovery rate of the nontoxic OxyS mutant is between the rate of the vector control and OxyS, not similar to that of the control cells as stated in the text.

Reply: Indeed so, we corrected the text accordingly

3) Page 9: The authors should rationalize why they used ZapA-GFP instead of FtsZ-GFP. Presumably it is because they were concerned that expression of *ftsZ-gfp* might suppress the effects of Kil. However, *zapA-gfp* expression may have the same effect to some extent, as it can bundle FtsZ polymers and protect against FtsZ assembly inhibitors. Did the authors try immunofluorescent staining using anti-FtsZ to avoid perturbing levels of cell division proteins?

Reply: *zapA-gfp* fusion is located within the native locus of *zapA*. We used *zapA-gfp* and not *ftsZ-gfp* fusion because FtsZ function cannot be fully replaced by the chromosomal

ftsZ-gfp (Thanedar and Margolin, Current Biology. 2004; Ben Yehuda and Losick, Cell .2002). Since no plasmids were used we expected no *trans* expression effects.

4) In Fig. 4A, it looks like ZapA-GFP forms fairly coherent structures at somewhat regular intervals. Assuming that these reflect FtsZ, it is not necessarily the case that "ring assembly is impaired", stated on page 9, although it is clear that the structures cannot progress further in cytokinesis.

Reply: Inadvertently, we carried out the experiments presented in former Fig. 4A at 30°C, a suboptimal growth temperature. We repeated these experiments at 37°C and found that under optimal growth conditions, in cells expressing toxic OxyS, *zapA-gfp* is diffused, forming no structures, indicating that ring assembly is indeed impaired. The revised manuscript includes a new A section in Fig 5.

5) In Fig. 4B, a significant fraction of cells are still filamentous after oxyS expression even in the absence of *kil*. Does that implicate alternative Kil-like proteins such as DicB, which may also be induced by OxyS?

Reply: About 10% of *kil* mutant cells expressing OxyS are filamentous vs. 60% of wild type cells that express OxyS. As also suggested by this referee; since Δkil cell filamentation seems scarce and at random, we assume that the *oxyS*-dependent decrease in *nusG* levels leading to continues transcription could induce alternative *kil*-like phage proteins.

6) Fig. 4C: why does *kil* expression decrease more than 2-fold in the first 15 minutes after peroxide addition?

Reply: We speculate that this initial decrease reflects an effect of hydrogen peroxide on cell metabolism. It is worth noting that *kilR* level in wild-type cells increases with time above background, while *oxyS* mutant *kilR* remains below background level.

7) MC4100 already has a *relA1* allele, so why was it necessary to make a *relA::kan* or *relA::cat* derivative?

Reply: All experiments were carried out in MG1655 strain that is considered wild type. MC4100 *relA1* mutant carries an insertion and the mutation cannot be moved.

8) MC4100 also has a *spoT1* mutation (Spira, Hu & Ferenci, Microbiology 2008) that should render it devoid of ppGpp. Is such a strain a valid model for a typical response to stresses?

Reply: Our experiments were carried out in MG1655 strain in which we deleted the *relA* allele only.

9) Whenever the pftsQAZ plasmid is used, the authors assume that the effects they see

are due to FtsZ. While this is likely true, they cannot rule out effects mediated by extra FtsA (or FtsQ?) as well, unless they express FtsZ by itself.

Reply: We agree with the referee, although it is likely true that the effect detected is due to FtsZ, we cannot rule out effects of FtsQA. We corrected the text accordingly. Page 10

10) Page 10, top, and Fig. 4C: as mentioned above, the population, albeit reduced, of significantly longer cells even in the absence of OxyS or Kil suggests that there are alternative pathways to help cells exposed to oxidative stress survive that involve cell division arrest.

Reply: Indeed so, hydrogen peroxide produces DNA damage that in turn induces the SOS response and thus the increase in cell length. Yet, cells expressing *oxyS* are significantly longer than *oxyS* mutant cells. In addition, we find that OxyS is toxic in SOS off cells indicating that the effect of OxyS is independent of SOS (see also new Appendix Fig. S6). Page 14

Minor English/typo issues:

- 1) Page 3 line 6: this sentence would be clearer after deleting the phrase "the expression of which is". Corrected
- 2) Page 5 line 14: change to "harmless clustered in two sites:" Corrected
- 3) Page 6 line 5: change to "targets whose putative complementary sites might match the changes." Corrected
- 4) Page 7 line 14: delete "a" Corrected
- 5) Page 8 line 3: delete comma Corrected
- 6) Page 8: there is inconsistent use of present and past tense, particularly in this paragraph. Corrected
- 7) Page 9, line 17: change to "fusions" Corrected
- 8) Page 13, lines 7-8: change to "we show that the middle hairpin B and the single-stranded region of OxyS interact..." Corrected
- 9) Fig 3C, y-axis should define it as "kil mRNA copy number" Corrected
- 10) Fig. 4C: y-axis: "Length" is misspelled. Corrected

We thank the referees for their comments and hope that the revised manuscript is now acceptable for publication.

Sincerely,

Shoshy Altuvia

Thank you for submitting your revised manuscript for our consideration. It has now been seen once more by all three original referees, and I am pleased to inform you that all of them are generally satisfied with your revisions. As you will see from the comments copied below, they only retain a number of presentational concerns related to both text and figures, which I would therefore like to invite you to address during a final round of minor revision.

Once we will have received the modified files addressing the various editorial and referee points, we should hopefully be able to swiftly proceed with final acceptance and production of the manuscript. I look forward to receiving your final version!

 REFEREE REPORTS

Referee #1:

In the revised version of this manuscript, the authors have addressed most of my previous concerns. However, I have a few remaining comments, mainly regarding the presentation of the data:

- Figure 2A:

- o change "NusG" to "nusG" (italics)
- o add label "3'" to the 3' ends of oxyS and nusG
- o the positions of the green arrows (-16C, -15C) needs to be shifted further to the left.

- Figure 3:

- o Add labels "A" and "B"
- o change "NusG" to "nusG" (italics)
- o add label "3'" to the 3' ends of oxyS and nusG

- Figure 3:

- o Panel E: the 0 min. time-point should be set to 100% (numbers need to be added to y-axis)
- o Panel F, Western Blot: looking at the data it seems that NusG-SPA levels remain constant, while Hfq levels decrease. Therefore, another loading control might be required for this experiment.

- Figure 5

- o Panel C: italicize gene names

- Figure S1A: add error bars, how many times have these experiments been performed?

- Figure S2B: add concentrations of RNAs to the figure legend. Why is there no increase in RNA duplex formation when increasing amounts of OxyS are added to the reaction?

- Figure S4, legend: clarify "Cultures carrying Plac-oxyS were treated with 1 mM of IPTG at dilution". Also, looking at the data, it is hard to see a difference in the signal for tmRNA (is that what 'tm' stands for?) in the 10µg and 30µg samples.

- Figure S5, panel B: italicize gene names. Also, please clarify the title of this figure in the legend.

- Figure S7: italicize gene names (oxyS).

Referee #2:

In the present revised form, the authors have performed additional experiments which strengthened their model. The finding of this paper is of interest and provides a link between oxidative stress, cell growth inhibition, cell damage and reparation, and OxyS sRNA-dependent regulation.

Minor corrections.

- The labels A and B in Figure 3 are missing
- Figure S2B: The gel retardation assays showed that the binding of oxyS to nusG mRNA is not very efficient but I assume that the binding is probably highly dependent on Hfq. The concentration range of nusG mRNA is not written in the legend. The authors should also mentioned if they have used a full length mRNA or a fragment.

Referee #3:

The authors have done a good job in improving the manuscript. It is a nice and convincing story. I have a few minor comments below.

p. 10: The authors need to define ftsQAZ in the context of ftsZ. Also, my comment #9 suggesting that ftsQA effects might be involved was not addressed (I see no mention of this on p. 10, where the authors say that they corrected the text accordingly). The bottom of p. 11 mentions that ftsZ was overexpressed, but that is not strictly true.

Along the same lines, in the supplement, the source of the pFtsQAZ plasmid is cited as Peters et al. 2011. However, this plasmid is probably not a p15a vector, and Peters et al. does not mention this plasmid. Could the authors be referring to pTB63, a plasmid carrying ftsQAZ described in Bernhardt and de Boer 2004, which is cited in the main text? If so, this plasmid is a tetR derivative of pSC101, not pACYC, and is also not CmR as described in the plasmid list.

What does the asterisk after pSC101 in the plasmid list denote?

There is also no description of how the chromosomal zapA-GFP fusion from the strain described in Peters et al. was transferred into MG1655 to make A894.

2nd Revision - authors' response

13 November 2017

November 19, 2017

Dear Dr. Vodermaier,

Senior editor / The EMBO Journal

Re: Revised manuscript EMBOJ-2017-97651R1

Below is the point to point response to the referee's comments

Referee #1:

- Figure 2A:

o change "NusG" to "nusG" (italics)

o add label "3'" to the 3' ends of oxyS and nusG

o the positions of the green arrows (-16C, -15C) needs to be shifted further to the left.

All done

- Figure 3:

o Add labels "A" and "B"

o change "NusG" to "nusG" (italics)

o add label "3'" to the 3' ends of oxyS and nusG

Done

- Figure 3: Should be 4

o Panel E: the 0 min. time-point should be set to 100% (numbers need to be added to y-axis)

Done

o Panel F, Western Blot: looking at the data it seems that NusG-SPA levels remain constant, while Hfq levels decrease. Therefore, another loading control might be required for this experiment.

We used Hfq as a loading control *per se*, not because we assumed that its levels should not change under any circumstances. Moreover, using antibodies generated against Fis protein for the use of Fis levels as a loading control demonstrated that in *oxyS* mutant, Fis protein levels decrease, similarly to Hfq (see below). Also, the levels of at least three proteins detected using anti Hfq antibodies (either hfq multimers or proteins recognized by non-specific cross reactions) decreased, similar to Hfq.

- Figure 5

o Panel C: italicize gene names

Done

- Figure S1A: add error bars, how many times have these experiments been performed?
Fig S1A now include error bars. As stated in the legends "Results are displayed as mean of 2 biological experiments \pm standard deviation.

- Figure S2B: add concentrations of RNAs to the figure legend. Why is there no increase in RNA duplex formation when increasing amounts of OxyS are added to the reaction?
Legend to Fig S2B now includes concentrations and length of *nusG* fragment
We speculate that the discontinues complementarity between OxyS and NusG renders OxyS-NusG binding strongly dependent on Hfq. Still, a slight increase in binding upon addition of increasing amounts of OxyS can be detected.

- Figure S4, legend: clarify "Cultures carrying Plac-oxyS were treated with 1 mM of IPTG at dilution". Also, looking at the data, it is hard to see a difference in the signal for tmRNA (is that what 'tm' stands for?) in the 10 μ g and 30 μ g samples.

The legend of Fig. S4 was rephrased.

tmRNA-We suspect a slight loading inaccuracy with the 10 μ g sample. OxyS levels (chromosomal vs. plasmid encoded) were estimated based on the 30 μ g samples.

- Figure S5, panel B: italicize gene names. Also, please clarify the title of this figure in the legend.

The legend to Fig S5 was rephrased

- Figure S7: italicize gene names (oxyS).

Corrected

Referee #2:

Minor corrections.

- The labels A and B in Figure 3 are missing

Labels were added

- Figure S2B: The gel retardation assays showed that the binding of oxyS to nusG mRNA is not very efficient but I assume that the binding is probably highly dependent on Hfq. The concentration range of nusG mRNA is not written in the legend. The authors should also mentioned if they have used a full length mRNA or a fragment.

Legend to Fig S2B now includes concentrations and length of *nusG* fragment

Referee #3:

The authors have done a good job in improving the manuscript. It is a nice and convincing story. I have a few minor comments below.

p. 10: The authors need to define *ftsQAZ* in the context of *ftsZ*. Also, my comment #9 suggesting that *ftsQA* effects might be involved was not addressed (I see no mention of this on p. 10, where the authors say that they corrected the text accordingly). The bottom of p. 11 mentions that *ftsZ* was overexpressed, but that is not strictly true.

The sentence mentioned above was rephrased (p11) and we were careful with similar sentences. Also *ftsQAZ* was defined in the context of *ftsZ* (see page 10). We also changed *ftsZ* to *FtsQAZ* in the rif resistance table.

Along the same lines, in the supplement, the source of the pFtsQAZ plasmid is cited as Peters et al. 2011. However, this plasmid is probably not a p15a vector, and Peters et al. does not mention this plasmid. Could the authors be referring to pTB63, a plasmid carrying *ftsQAZ* described in Bernhardt and de Boer 2004, which is cited in the main text? If so, this plasmid is a tetR derivative of pSC101, not pACYC, and is also not CmR as described in the plasmid list.

Indeed, the plasmid was cited correctly in the text but not in the list of strains. I corrected its citation in the table of strains (I am grateful for this comment).

What does the asterisk after pSC101 in the plasmid list denote?

pSC101* is a single copy plasmid based on pSC101 origin. Referenced in the table of strains.

There is also no description of how the chromosomal zapA-GFP fusion from the strain described in Peters et al. was transferred into MG1655 to make A894.

The table of strains mention P1 transduction as a way of transferring chromosomal fusions and mutations.

I thank you and hope that the revised manuscript is now acceptable for publication.

Sincerely,

Shoshy Altuvia

Corresponding Author Name: Shoshy Altuvia; Wolfgang R Hess

Manuscript Number: EMBOJ-2017-97651